# EXPLORING DOMAIN SHIFT WITH DIFFUSION-BASED ADAPTATION FOR REAL IMAGE DEHAZING

## ABSTRACT

Conventional supervised single-image dehazing methods, which are trained with substantial synthetic hazy-clean image pairs, have achieved promising performance. However, they often fail to tackle out-of-distribution hazy images, due to the domain shift between source and target scenarios (e.g., between indoor and outdoor, between synthetic and real). In this work, we observe the opportunity for improving such dehazing models' generalization ability without modifying the architectures or weights of conventional models by adopting the diffusion model to transfer the distribution of input images from target domain to source domain. Specifically, we train a denoising diffusion probabilistic model (DDPM) with source hazy images to capture prior probability distribution of the source domain. Then, during the test-time the obtained DDPM can adapt target hazy inputs to source domain in the reverse process from the perspective of conditional generation. The adapted inputs are fed into a certain state-of-the-art (SOTA) dehazing model pre-trained on source domain to predict the haze-free outputs. Note that, the whole proposed pipeline, termed **Diff**usion-based **AD**aptation (DiffAD), is model-agnostic and plug-and-play. Besides, to enhance the efficiency in real image dehazing, we further employ the predicted haze-free outputs as the pseudo labels to fine-tune the underlying model. Extensive experimental results demonstrate that our DiffAD is effective, achieving superior performance against SOTA dehazing methods in domain-shift scenarios.

## 1 INTRODUCTION

Hazy images often suffer from low contrast, poor visibility, and color distortion (Tan, 2008), imposing a negative impact on the downstream high-level vision tasks, such as object detection, image classification, and semantic segmentation. According to the atmospheric scattering model (ASM) (Narasimhan & Nayar, 2002; 2003), the hazing process is commonly formulated as:

$$I(x) = J(x)t(x) + A(1 - t(x)), \tag{1}$$

where $I(x)$ is the observed hazy image and $J(x)$ denotes the clean image of the same scene. $A$ and $t(x)$ are the global atmospheric light and the transmission map, respectively.

With the advancement of deep learning, various methods have been proposed to solve this highly ill-posed problem (Wu et al., 2021; Chen et al., 2024). Among them, well-designed architectures based on convolutional neural networks (CNNs) or transformers try to learn the dehazing priors from large-scale synthetic hazy-clean pairs and reach state-of-the-art (SOTA) performance. Such dehazing priors are particularly effective for synthetic hazy images with similar distribution to training data. However, the domain shift caused by different scenarios (indoor and outdoor) or different haze modes (synthetic and real) makes it challenging to generalize the learned dehazing priors from one specific domain (i.e., source) to another. For example, a model trained on indoor datasets fails to achieve desirable results in outdoor scenes as shown in Fig. 1. Previous methods (Shao et al., 2020; Chen et al., 2021; Yang et al., 2022; Yu et al., 2022) attempt to bridge this domain gap via generative adversarial networks (GANs) or unsupervised losses. These methods struggle to produce visually pleasing results. On one hand, GANs are difficult to train and may generate artifacts in the results. On the other hand, when the handcrafted priors (that unsupervised losses rely on) fail, the dehazing results may be biased. Moreover, these methods achieve domain adaptation by updating parameters

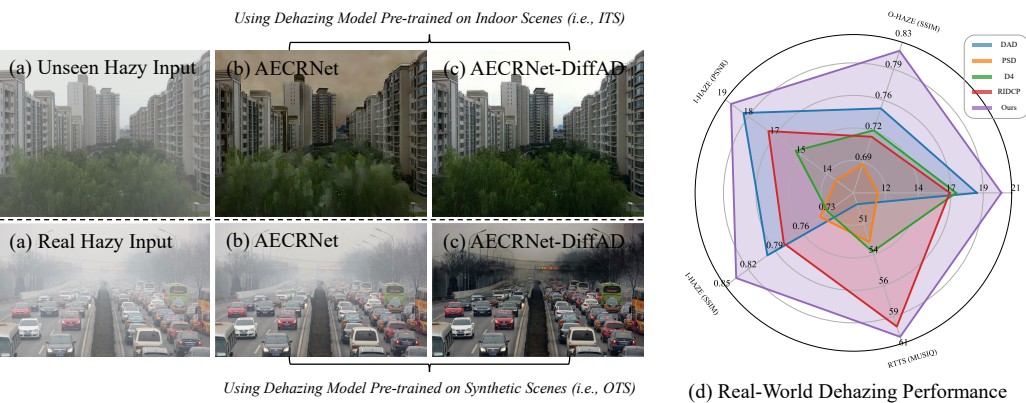

Figure 1: Left top: a model trained on indoor datasets fails to achieve desirable results in outdoor scenes, Left bottom: a model trained on synthetic datasets fails to achieve desirable results in real hazy scenes, Right: our DiffAD-FT outperforms SOTA dehazing models in real-world scenes.

during the training phase, which to some extent undermines the dehazing priors learned from the source domain. These dehazing priors have not received sufficient and deserved attention. *How to effectively leverage such dehazing priors in another unseen domain (biased from source domain)* remains unexplored.

Recently, diffusion probabilistic models (DDPM) (Sohl-Dickstein et al., 2015; Ho et al., 2020) have gradually surpassed GANs and exhibited great success in various tasks, such as image generation (Dhariwal & Nichol, 2021), image editing (Meng et al., 2022) and image restoration (Fei et al., 2023; Özdenizci & Legenstein, 2023). Given the powerful capacity for modeling complicated data distributions and generating high-quality images, DDPM can achieve domain translation by adding Gaussian noise and then gradually denoising (Meng et al., 2022; Su et al., 2022; Peng et al., 2023). Such property inspires a new research direction for effectively leveraging the learned dehazing prior in another unseen domain. One intuitive and feasible idea is to project hazy images from the target domain to the source domain by DDPM, and then perform dehazing through the dehazing model trained on the source domain. Since the weights are frozen after training, the dehazing priors encapsulated in the dehazing model remain intact and can be fully leveraged. However, as a kind of generative model, DDPM tends to slightly alter the image content during the domain translation process, introducing cumulative errors into the subsequent dehazing model. In addition, another drawback lies in the efficiency problem. *How to maintain the fidelity after domain translation* and *how to enhance the efficiency* are key challenges existing in this idea.

Based on the above discussions, we propose a novel **Diff**usion-based **AD**aptation paradigm (i.e., DiffAD) to explore the domain shift problem in image dehazing. DiffAD acts on the input hazy images to adjust the distribution. First, we train a DDPM with source hazy images to capture the prior probability distribution of the source domain. A source-Gaussian-source loop is built in this step and given a hazy image from the target domain (e.g., real-captured hazy image), we can adjust the distribution to make it align with the source domain (e.g., synthetic hazy image). A novel loss function is designed by considering the fidelity and quality to guide the generation during the reverse process (preventing the generative output from structure distortion and color variation). Then, the adapted hazy image can be directly fed into a certain SOTA dehazing model (e.g., AECRNet (Wu et al., 2021), Dehazeformer (Song et al., 2023), and FocalNet (Cui et al., 2023)) pre-trained on source domain to predict the haze-free output. The SOTA dehazing model is used for inference and will not change its weights and architecture. Therefore, the crucial dehazing priors can be fully explored and exploited. Finally, due to the absence of clean ground-truth images from target domain, we employ the haze-free outputs from our DiffAD as pseudo labels (with some necessary modifications) to fine-tune the underlying SOTA model. Surprisingly, the fine-tuned model no longer requires the DDPM, leading to a significant improvement in efficiency. In summary, our main contributions are as follows:

- We propose a novel Diffusion-based ADaptation paradigm (i.e., DiffAD) to explore the domain shift problem in image dehazing. To the best of our knowledge, this is the first time that the diffusion model has been employed to transfer the probability distribution of target domain (e.g., real-world hazy) into the source domain (e.g., synthetic hazy). DiffAD is a plug-and-play module that acts on the input image, thus will not alter the underlying dehazing model. The dehazing priors encapsulated in the underlying dehazing model can be fully explored and exploited.

- To guide the generation during the reverse process, a novel loss function is devised from the perspective of fidelity and quality. We show that the fidelity item can avoid information loss and the quality item brings controllability, ensuring the generation of high-quality haze-free images.

- We further take the obtained haze-free images as the pseudo labels to fine-tune the underlying dehazing model. This updated model can be directly applied to recover real-world hazy images with enhanced efficiency.

## 2 RELATED WORK

**Single Image Dehazing.** Early efforts (Fattal, 2008; Tan, 2008; He et al., 2010; Fattal, 2014; Zhu et al., 2015; Berman et al., 2016) made in image dehazing relies on ASM and primarily focus on handcraft priors observed from both hazy and haze-free images. These methods achieve promising results but fail in scenes that do not satisfy their assumptions. The advent of deep learning has revolutionized image dehazing by freeing it from handcrafted priors. A variety meticulously designed architectures (Cai et al., 2016; Ren et al., 2016; Li et al., 2017; Zhang & Patel, 2018; Liu et al., 2019; Dong et al., 2020; Dong & Pan, 2020; Qin et al., 2020; Wu et al., 2021; Guo et al., 2022; Hong et al., 2022; Ye et al., 2022; Song et al., 2023; Zheng et al., 2023; He et al., 2023; Chen et al., 2024; Zhang et al., 2024) has been proposed to learn image dehazing from the large-scale synthetic datasets (Li et al., 2018; Liu et al., 2021). For example, Qin et al. (2020) introduce attention mechanisms to CNNs and significantly improve the dehazing performance. Song et al. (2023) propose a transformer-based architecture to further promote image dehazing. Although these learning-based methods achieve impressive results, they tend to over-fit the training set and demonstrate poor generalization ability on unseen hazy images.

**Domain Adaptation for Image Dehazing.** To address the domain shift when encountering unseen hazy images, some studies (Li et al., 2019; Shao et al., 2020; Chen et al., 2021; Yu et al., 2022; Li et al., 2022) attempt to improve the generalization ability of dehazing models through domain adaptation. For instance, a representative solution (Shao et al., 2020; Li et al., 2022) involves utilizing GANs to perform translation between the source and the target domain, followed by adapting model to the target domain. (Li et al., 2019; Chen et al., 2021; Yu et al., 2022) start from physical priors and adapt the dehazing model to the target domain in an unsupervised manner. However, due to the updating of model parameters, these methods struggle to preserve the well-learned dehazing priors from large-scale synthetic datasets.

**Diffusion models.** Recently, denoising diffusion probabilistic models (DDPMs) (Sohl-Dickstein et al., 2015; Ho et al., 2020) have attracted widespread attention from researchers as a type of generative model. DDPM gradually converts simple Gaussian noise to complex data distribution by a Markov chain. Many studies have demonstrated the superiority of DDPM across various tasks (Dhariwal & Nichol, 2021; Vahdat et al., 2021; Yin et al., 2022; Su et al., 2022; Meng et al., 2022; Gao et al., 2022; Fei et al., 2023; Özdenizci & Legenstein, 2023; Peng et al., 2023). In image dehazing, a prevalent way to utilize DDPM is mapping the hazy image to the clear one in a conditional manner (Özdenizci & Legenstein, 2023; Yu et al., 2023; Wang et al., 2024). Different from previous works, in this paper, we employ DDPM to project the hazy image from the target to the source domain, aiming to preserve well-learned dehazing priors of the source domain.

## 3 PRELIMINARY

Denoising Diffusion Probabilistic Model (DDPM) (Sohl-Dickstein et al., 2015; Ho et al., 2020) is a kind of generative models that transforms back and forth between complex data distribution and

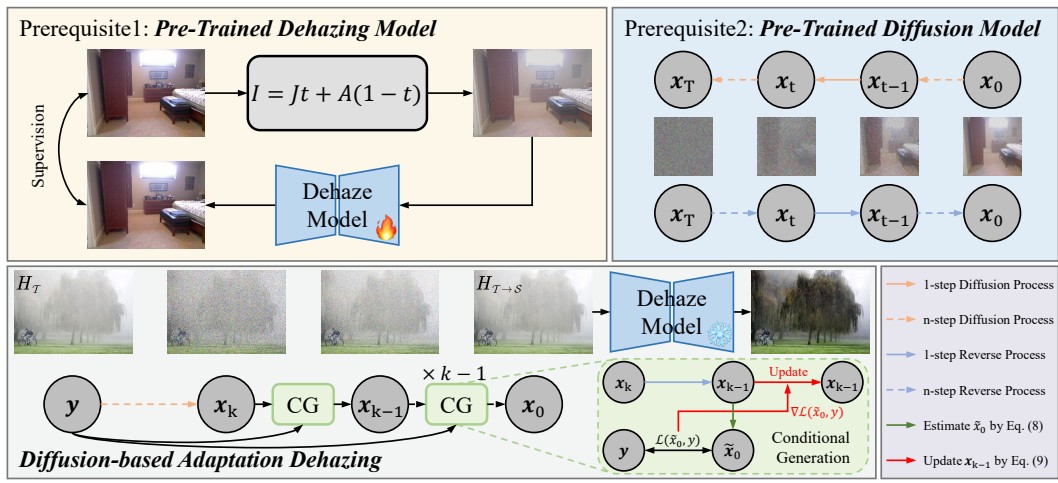

Figure 2: Overall pipeline of **Diff**usion-based **AD**aptation paradigm (DiffAD). It contains three steps: pre-trained dehazing model, pre-trained diffusion model, and diffusion-based adaptation dehazing.

simple Gaussian distribution. A DDPM mainly consists of two processes: the *diffusion process* and the *reverse process*.

In the diffusion process, the data $x_0$ is progressively corrupted by the injection of a slight amount of Gaussian noise over $T$ time steps, transforming into $x_T$:

$$q\left(x_{1:T} \mid x_0\right) = \prod_{t=1}^{T} q\left(x_t \mid x_{t-1}\right), \quad q\left(x_t \mid x_{t-1}\right) = \mathcal{N}\left(x_t; \sqrt{1-\beta_t}x_{t-1}, \beta_t \mathbf{I}\right), \quad (2)$$

where $t$ and $\beta_{1:T}$ denotes diffusion step and predefined variance schedule, respectively. Let $\alpha_t = 1 - \beta_t$, an intermediate $x_t$ can be sampled directly from $x_0$:

$$q\left(x_t \mid x_0\right) = \mathcal{N}\left(x_t; \sqrt{\bar{\alpha}_t}x_0, \left(1-\bar{\alpha}_t\right)\mathbf{I}\right), \quad (3)$$

where $\bar{\alpha}_t = \prod_{i=1}^{t} \alpha_i$ and $\epsilon \sim \mathcal{N}(0, \mathbf{I})$. The corresponding closed form can be written as:

$$x_t = \sqrt{\bar{\alpha}_t}x_0 + \sqrt{1-\bar{\alpha}_t}\epsilon. \quad (4)$$

Contrary to the diffusion process, the reverse process starts from a Gaussian noise $x_T$, aiming to recover data $x_0$ by denoising gradually:

$$p_\theta\left(x_{0:T-1} \mid x_T\right) = \prod_{t=1}^{T} p_\theta\left(x_{t-1} \mid x_t\right), \quad p_\theta\left(x_{t-1} \mid x_t\right) = \mathcal{N}\left(x_{t-1}; \mu_\theta\left(x_t, t\right), \Sigma_\theta\mathbf{I}\right), \quad (5)$$

where $\Sigma_\theta$ is the predefined (Ho et al., 2020) or learnable (Nichol & Dhariwal, 2021) variance. $\mu_\theta\left(x_t, t\right)$ is the mean, which can be derived by applying the reparameterization technique:

$$\mu_\theta\left(x_t, t\right) = \frac{1}{\sqrt{1-\beta_t}}\left(x_t - \frac{\beta_t}{\sqrt{1-\bar{\alpha}_t}}\epsilon_\theta\left(x_t, t\right)\right), \quad (6)$$

where $\epsilon_\theta$ is a noise estimator, typically adopting U-Net (Ronneberger et al., 2015) as its architecture. The training objective of DDPM is to enable $\mu_\theta$ to accurately estimate the noise of arbitrary intermediate image $x_t$:

$$L_{DDPM} = \left\|\epsilon_\theta\left(x_t, t\right) - \epsilon\right\|^2. \quad (7)$$

## 4 METHODOLOGY

### 4.1 DIFFAD PIPELINE

We try to explore the domain shift problem for image dehazing. The detailed definition is as follows: given a source domain $\mathcal{S} = \{H_{\mathcal{S}_i}, C_{\mathcal{S}_i}\}_{i=1}^{N_\mathcal{S}}$ comprising $N_\mathcal{S}$ source hazy images $H_{\mathcal{S}_i}$ and corresponding source clear labels $C_{\mathcal{S}_i}$, along with the dehazing model $\Phi$ that properly learns dehazing priors from $\mathcal{S}$, we aim to improve the generality of $\Phi$ in target domain $\mathcal{T} = \{H_{\mathcal{T}_i}\}_{i=1}^{N_\mathcal{T}}$ (which only contains $N_\mathcal{T}$ unlabeled target hazy images $H_{\mathcal{T}_i}$).

Previous methods adapt models to the target domain $\mathcal{T}$ (Chen et al., 2021; Yu et al., 2022). However, they neglect the useful dehazing priors encoded in $\Phi$ learned from the source domain (e.g., large-scale synthetic datasets). On the contrary, we propose a novel framework called **Diff**usion-based **AD**aptation (DiffAD) to perform input adaptation rather than model adaptation. The key idea of the proposed DiffAD is to project the target hazy image $H_\mathcal{T}$ to the source domain $\mathcal{S}$ by a controllable diffusion model.

The whole pipeline is illustrated in Fig. 2. To start with, we choose a well-designed dehazing model $\Phi$ pre-trained on the source domain $\mathcal{S}$ with the dehazing priors already encoded. Then, we train a standard unconditional DDPM to capture the complicated data distribution on source hazy images $\{H_{\mathcal{S}_i}\}_{i=1}^{N_\mathcal{S}}$ by optimizing equation 7. With the trained DDPM, we are able to perform input adaptation during test-time. Given a hazy image $H_\mathcal{T}$ from the target domain $\mathcal{T}$ (e.g., real-world hazy image), we project it to the source domain $\mathcal{S}$ (e.g., synthetic hazy image), denoted as $H_{\mathcal{T}\to\mathcal{S}}$, by adding noise to $H_\mathcal{T}$ and going through the reverse process. More details can be found in Sec. 4.1.1 and Sec. 4.1.2. Finally, we dehaze the projected image $H_{\mathcal{T}\to\mathcal{S}}$ by pre-trained dehazing model $\Phi$ with well-learned dehazing priors.

### 4.1.1 CONDITIONAL GENERATION

Although aligning $H_\mathcal{T}$ with the source domain $\mathcal{S}$ can revitalize the well-learned dehazing priors, content changes are inevitable in the unconditional reverse process due to its generation nature. As shown in Fig. 3 (c), aligning $H_\mathcal{T}$ with $\mathcal{S}$ in an unconditional manner enables FocalNet (Cui et al., 2023) to properly leverage learned dehazing priors. However, as indicated by the red box of Fig. 3 (c), structural deformation and color distortion are introduced. Thus, directly recover the diffused image through equation 5 is sub-optimal.

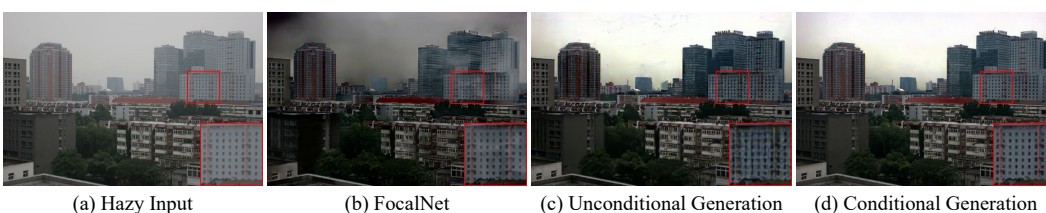

| (a) Hazy Input | (b) FocalNet | (c) Unconditional Generation | (d) Conditional Generation |

Figure 3: (a) a hazy input, (b) dehazing result by FocalNet, (c) dehazing result by DiffAD with unconditional generation, and (d) dehazing result by DiffAD with conditional generation.

Inspired by Dhariwal & Nichol (2021); Fei et al. (2023), we can introduce the custom loss function $\mathcal{L}(x_t, y)$ to control the reverse process towards the condition $y$ at each time step $t$. The conditional generation can be achieved by shifting the mean of unconditional distribution $\mu_\theta(x_t, t)$ in equation 5 by $g\Sigma_\theta \nabla_{x_t} \mathcal{L}(x_t, y)$, where $g$ is a scaling factor controlling the magnitude of guidance. In our DiffAD, we use the hazy input $H_\mathcal{T}$ as the condition $y$, since we aim to achieve higher fidelity by constraining the projected $H_{\mathcal{T}\to\mathcal{S}}$ to have similar structure and color distribution to $H_\mathcal{T}$. Following Fei et al. (2023), to eliminate the impact from noise, we replace $x_t$ with $\tilde{x}_0$ (the guidance is conditional on $\tilde{x}_0$), which is noise-free and can be predicted directly from $x_t$ at each time step $t$ based on equation 4:

$$\tilde{x}_0 = \frac{x_t}{\sqrt{\bar{\alpha}_t}} - \frac{\sqrt{1 - \bar{\alpha}_t}}{\sqrt{\bar{\alpha}_t}}\epsilon, \tag{8}$$

We omit the time step $t$ in $\tilde{x}_0$ for simplification. In this way, equation 5 can be rewritten as:

$$p_\theta(x_{t-1} \mid x_t, y) = \mathcal{N}(x_{t-1}; \mu_\theta(x_t, t) + g\Sigma_\theta \nabla_{x_t} \mathcal{L}(\tilde{x}_0, y), \Sigma_\theta \mathbf{I}), \tag{9}$$

### 4.1.2 CUSTOM LOSS FUNCTION

The loss function $\mathcal{L}(\tilde{x}_0, y)$ works in **test time** by guiding the projection from $H_{\mathcal{T}}$ to $H_{\mathcal{T} \to \mathcal{S}}$ in terms of fidelity and quality. Accordingly, the total loss can be divided into two items: fidelity loss $\mathcal{F}(\tilde{x}_0, y)$ and quality loss $\mathcal{Q}(\tilde{x}_0, y)$. The former contains a spatial consistency loss $\mathcal{L}_{sc}$ and a color consistency loss $\mathcal{L}_{cc}$. The latter contains a white balance loss $\mathcal{L}_{wb}$ and a region-aware DCP loss $\mathcal{L}_{rdcp}$.

**Fidelity Loss.** We design the fidelity loss $\mathcal{F}(\tilde{x}_0, y)$ from the perspective of preventing both structure deformation and color distortion to ensure the fidelity of the projected image $H_{\mathcal{T} \to \mathcal{S}}$. In general circumstances, we don't need to consider the issue of fidelity, since constraints have been imposed by image distance losses (e.g., MSE). However, in our DiffAD pipeline, MSE may fail the image adaptation ($\tilde{x}_0$ and $y$ should exhibit distinct distributions). Thus, we adopt the spatial consistency loss $\mathcal{L}_{sc}$ from Guo et al. (2020), which encourages spatial coherence of $H_{\mathcal{T} \to \mathcal{S}}$ through preserving the structural gradient (rather than intensity) between $\tilde{x}_0$ and $y$:

$$\mathcal{L}_{sc} = \frac{1}{N} \sum_{i=1}^{N} \sum_{j \in \Omega(i)} (|\tilde{x}_0^i - \tilde{x}_0^j| - |y^i - y^j|)^2, \tag{10}$$

where $N$ denotes the number of pixels, $\Omega(i)$ represents the four adjacent pixels (top, down, left and right) centered at the pixel $i$. Similarly, a color consistency loss $\mathcal{L}_{cc}$ is designed to encourage color coherence of $H_{\mathcal{T} \to \mathcal{S}}$ through preserving the relative color (between channels) between $\tilde{x}_0$ and $y$.

$$\mathcal{L}_{cc} = \frac{1}{N} \sum_{i=1}^{N} \sum_{\forall (j,k) \in \varepsilon} (|\tilde{x}_0^{i,j} - \tilde{x}_0^{i,k}| - |y^{i,j} - y^{i,k}|)^2, \varepsilon = \{(R,G), (R,B), (G,B)\}, \tag{11}$$

where $\varepsilon$ denotes the color channel pairs [1]. To the best of our knowledge, this is the first time that color consistency loss $\mathcal{L}_{cc}$ is proposed to align the color information. The fidelity loss can be formulated as the weighted sum of $\mathcal{L}_{sc}$ and $\mathcal{L}_{cc}$:

$$\mathcal{F}(\tilde{x}_0, y) = \lambda_{sc} \mathcal{L}_{sc} + \lambda_{cc} \mathcal{L}_{cc}, \tag{12}$$

where $\lambda_{sc}$ and $\lambda_{cc}$ are weight coefficients.

**Quality Loss.** In addition to fidelity loss, we propose the controllable quality loss $\mathcal{Q}(\tilde{x}_0, y)$ that users can adjust white balanced effect and extent of dehazing. For varicolored hazy scenes, we revise the color constancy loss from Guo et al. (2020) and re-name it to white balance loss $\mathcal{L}_{wb}$. It eliminates the color cast of $\tilde{x}_0$ based on the Gray-World Assumption (Buchsbaum, 1980). According to equation 1, regions with dense haze demonstrate increased sensitivity to atmospheric light with color shift. Therefore, we introduce haze density $\mathcal{D}(y)$ estimated by dark channel prior (DCP) (He et al., 2010) as the spatial weights. The white balance loss $\mathcal{L}_{wb}$ can be formulated as:

$$\mathcal{L}_{wb} = \sum_{\forall (i,j) \in \varepsilon} \left( \mu^i(\mathcal{D}(y) \cdot \tilde{x}_0) - \mu^j(\mathcal{D}(y) \cdot \tilde{x}_0) \right)^2, \varepsilon = \{(R,G), (R,B), (G,B)\}, \tag{13}$$

where $\mu(\cdot) \in \mathbb{R}^C$ is the mean value computed across spatial dimensions for each color channel. Our $\mathcal{L}_{wb}$ can be regarded as the enhanced version of the color constancy loss.

DCP loss (Golts et al., 2020; Li et al., 2020) is widely used in real image dehazing. However, DCP tends to fail in the sky region (He et al., 2010). We revise the original DCP loss (Li et al., 2020) and re-name it to region-aware DCP loss $\mathcal{L}_{rdcp}$. Accordingly, we exclude the sky region with a mask $\mathcal{M}_{sky}$ generated by Zou et al. (2022) to avoid potential inaccurate calculation of DCP. The $\mathcal{L}_{rdcp}$ is optimized over $z = \Phi(\tilde{x}_0)$, and we employ $\mathcal{D}(z)$ as the spatial weights. We formulate $\mathcal{L}_{rdcp}$ as:

$$\mathcal{L}_{rdcp} = \mathcal{M}_{sky} \cdot \mathcal{D}(z) \cdot \mathcal{J}(z), \tag{14}$$

where $\mathcal{J}(\cdot)$ denotes the original DCP loss (Li et al., 2020). The quality loss can be formulated as:

$$\mathcal{Q}(\tilde{x}_0, y) = \lambda_{wb} \mathcal{L}_{wb} + \lambda_{dcp} \mathcal{L}_{rdcp}, \tag{15}$$

where $\lambda_{wb}$ and $\lambda_{dcp}$ are weight coefficients which are adjustable (**refer to supplemental material**).

**Total Loss.** The total loss $\mathcal{L}(\tilde{x}_0, y)$ can be formulated by combining fidelity loss and quality loss:

$$\mathcal{L}(\tilde{x}_0, y) = \mathcal{F}(\tilde{x}_0, y) + \mathcal{Q}(\tilde{x}_0, y) \tag{16}$$

---

[1]Both of the spatial consistency loss and the color consistency loss can be calculated on local regions.

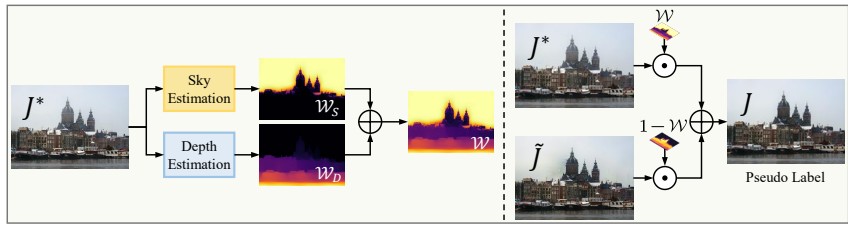

Figure 4: Overview of our fine-tune pipeline. We design a fully automatic pipeline to generate the pseudo labels for fine-tuning the underlying dehazing model.

### 4.2 DIFFAD FOR REAL IMAGE DEHAZING

Due to the difficulty in obtaining a large-scale hazy-clean image pairs under real-world scenarios, improving the model's (pre-trained on synthetic hazy-clean pairs) generalization ability is a promising research direction. A typical application of our DiffAD is to remove the haze of real-captured images, which are unlabeled. Although DiffAD provides an effective solution for domain shift problem in real image dehazing, it is highly time-consuming due to the iterative reverse process. To enhance the efficiency, we collect some real hazy images and generate corresponding high-quality pseudo labels with a pre-trained SOTA dehazing model $\Phi$ and our DiffAD.

As illustrated in Fig. 4, we design a fully automatic pipeline to generate the pseudo label $J$. The output $J^*$ of the pre-trained dehazing model $\Phi$ is also embedded to avoid catastrophic forgetting. Specifically, a high-quality pseudo label must satisfy simultaneously with **(a) visibility within dense haze regions, and (b) artifact-free**. Benefiting from the controllable nature of our DiffAD, we can easily obtain pseudo label $\tilde{J}$ that satisfy property (a) by adopting relatively larger $\lambda_{dcp}$. To further fulfill the property (b), we find the the output $J^*$ of the pre-trained dehazing model $\Phi$ quite fits. As illustrated in Fig. 5, we first compute the weight map $\mathcal{W}$ by adding the sky mask $\mathcal{W}_S$ and depth map $\mathcal{W}_D$ estimated from $J^*$. Then, the $\mathcal{W}$ is utilized to fuse $\tilde{J}$ and $J^*$ in a weighted addition manner to generate the refined pseudo label $J$. In our implementation, the methods described in (Zou et al., 2022) and (Yang et al., 2024) are adopted to estimate $\mathcal{W}_S$ and $\mathcal{W}_D$, respectively.

Figure 5: The refine process used in high-quality pseudo label generation.

Finally, the underlying dehazing model $\Phi$ is fine-tuned with generated pseudo labels. A depth estimation module is added into the original architecture and the depth information is integrated via SFT layers (Wang et al., 2018) into the encoder for better performance. **Please refer to our supplemental material for more details.**

## 5 EXPERIMENTS

### 5.1 CAN DIFFAD RELIEVE THE DOMAIN SHIFT ISSUE?

Here, we consider two common types of domain shift: (1) between different scene types: apply a model pre-trained on indoor/outdoor data to outdoor/indoor scenes, (2) between different haze types: apply a model pre-trained on synthetic data to real-captured scenes.

Table 1: The performance of scene type adaptation of our DiffAD pipeline.

|  | OTS / SOTS-indoor | | ITS / SOTS-outdoor | |
| --- | --- | --- | --- | --- |
|  | PSNR↑ | SSIM↑ | PSNR↑ | SSIM↑ |
| (CVPR'21) AECRNet | 22.40 | 0.9097 | 17.08 | 0.8475 |
| (Ours) AECRNet-DiffAD | **25.05** | **0.9224** | **20.64** | **0.8759** |
| (TIP'23) Dehazeformer | 24.07 | 0.9317 | 20.67 | 0.8827 |
| (Ours) Dehazeformer-DiffAD | **26.23** | **0.9356** | **23.88** | **0.9191** |
| (ICCV'23) FocalNet | 17.10 | 0.8280 | 19.81 | 0.8582 |
| (Ours) FocalNet-DiffAD | **24.84** | **0.9291** | **21.07** | **0.8865** |

Table 2: The performance of haze type adaptation of our DiffAD pipeline.

|  | ITS / I-HAZE | | OTS / O-HAZE | | Wu / O-HAZE | |
| --- | --- | --- | --- | --- | --- | --- |
|  | PSNR↑ | SSIM↑ | PSNR↑ | SSIM↑ | PSNR↑ | SSIM↑ |
| (CVPR'21) AECRNet | 11.34 | 0.5515 | 16.48 | 0.6979 | 17.23 | 0.7637 |
| (Ours) AECRNet-DiffAD | **13.36** | **0.6580** | **17.71** | **0.7317** | **19.11** | **0.7911** |
| (TIP'23) Dehazeformer | 12.60 | 0.6078 | 16.38 | 0.6959 | 17.09 | 0.7693 |
| (Ours) Dehazeformer-DiffAD | **14.30** | **0.7216** | **17.94** | **0.7276** | **19.11** | **0.7979** |
| (ICCV'23) FocalNet | 10.95 | 0.4870 | 16.82 | 0.7136 | 18.01 | 0.7870 |
| (Ours) FocalNet-DiffAD | **13.70** | **0.6786** | **18.09** | **0.7424** | **19.28** | **0.7957** |

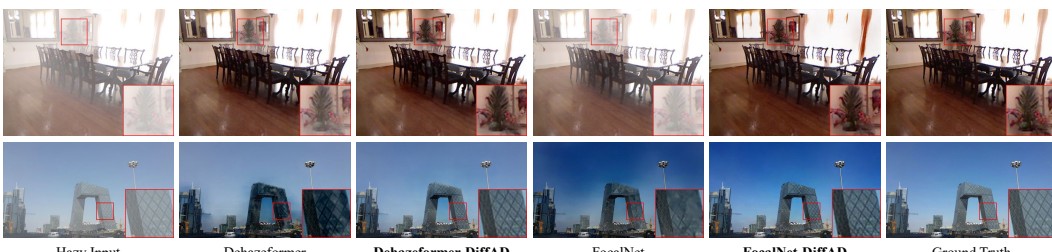

| Hazy Input | Dehazeformer | **Dehazeformer-DiffAD** | FocalNet | **FocalNet-DiffAD** | Ground Truth |

Figure 6: Top: qualitative result under "OTS / SOTS-indoor" setting, Bottom: qualitative result under "ITS / SOTS-outdoor" setting.

**Datasets and Evaluation Metrics.** We employ three widely-used synthetic datasets as the source domains, including ITS dataset (Li et al., 2018), OTS dataset (Li et al., 2018) and Wu's dataset (Wu et al., 2023). For domain shift (1), two synthetic datasets are selected for quantitative assessment: SOTS-indoor dataset (Li et al., 2018), SOTS-outdoor dataset (Li et al., 2018). For domain shift (2), we adopt O-HAZE (Ancuti et al., 2018a; Kar et al., 2021) and I-HAZE (Ancuti et al., 2018b; Kar et al., 2021) datasets as target real domains. In addition, we adopt PSNR and SSIM as evaluation metrics.

**Implementation Details.** As illustrated in Fig. 2, DiffAD contains three main steps. For step one, We select AECRNet (Wu et al., 2021), Dehazeformer (Song et al., 2023), and FocalNet (Cui et al., 2023) as our base networks. We re-train their models on source domains with public codes and default settings if their pre-trained models are not available. In step two, we train three denoising diffusion probabilistic models (DDPMs) from scratch on ITS (Li et al., 2018), OTS (Li et al., 2018) and Wu's dataset (Wu et al., 2023) (only the hazy images are employed for training). Each diffusion model is trained for 50k iterations using the Adam optimizer with $\beta_1 = 0.9$, $\beta_2 = 0.999$ and learning rate is set to $2e^{-5}$. We randomly crop images into $256 \times 256$ patches in the training phase. Following DDPM (Ho et al., 2020), we adopt linear noise schedule and set the number of diffusion steps as $T = 1000$. In step three, we empirically set the guidance scale $g$ to $0.8 \times HW$ for stable guidance, where $H$ and $W$ denote height and width of the input image, respectively. In the reverse process, we set $k = 10$, $\lambda_{sc} = 1$, $\lambda_{cc} = 0.1$, and $\lambda_{wb} = 1$ for all three DDPMs. $\lambda_{dcp}$ is used to control the extent of dehazing, and in our implementation, we set it to a fixed value (i.e., $5e^{-5}$).

**Scene Type Adaptation (between indoor and outdoor).** We evaluate the performance of scene type adaptation of our DiffAD between indoor and outdoor domains. Specifically, we choose the model pre-trained on OTS (source domain) to test the performance on SOTS-indoor (target domain). We denote this setting as "OTS / SOTS-indoor". "ITS / SOTS-outdoor" indicates the opposite setting. We equip our DiffAD with three selected state-of-the-art dehazing methods (i.e., AECRNet (Wu et al., 2021), Dehazeformer (Song et al., 2023), FocalNet (Cui et al., 2023)) to explore the domain shift between different scene types. The quantitative results are summarized in Table 1. It can be observed that previous methods tend to over-fit the source domain, resulting in poor generalization on the target domain with different scene types. Our method (labeled with **-DiffAD** suffix) can consistently enhance the generalization abilities of the selected models on the target domain. Especially, our DiffAD significantly enhance FocalNet's scene type adaptation performance on "OTS / SOTS-indoor" by achieving 7.74 dB and 0.1011 gains in terms of PSNR and SSIM. We also provide some qualitative results in Fig. 6.

**Haze Type Adaptation (between synthetic and real).** We also evaluate the performance of haze type adaptation of our DiffAD between synthetic and real domains. Specifically, we choose the model pre-trained on synthetic datasets to test the performance on real datasets. We denote this setting as "synthetic / real". We also equip our DiffAD with three selected SOTA dehazing methods to explore the domain shift between different haze types. The quantitative results are shown in Table 2. With the proposed DiffAD, selected models achieve robust performance improvements.

## 5.2 ABLATION STUDY

We also perform ablation study to verify the effectiveness of each component in $\mathcal{L}(\tilde{x}_0, y)$. We adopt AECRNet (Wu et al., 2021) as the underlying model, and measure PSNR and SSIM on scene type adaptation (i.e., "OTS / SOTS-indoor") and haze type adaptation (i.e., "Wu / O-HAZE").

Table 3 presents the results of different combinations of loss functions. Removing $\mathcal{L}_{sc}$ or $\mathcal{L}_{cc}$ or $\mathcal{L}_{dcp}$ causes performance drop in terms of PSNR and SSIM, demonstrating the effectiveness of $\mathcal{L}_{sc}$ and $\mathcal{L}_{cc}$ and $\mathcal{L}_{dcp}$. Due to the absence of varicolored scenes in SOTS-indoor dataset, we omit the ablation study of $\mathcal{L}_{wb}$ for "OTS / SOTS-indoor". When excluding $\mathcal{L}_{wb}$ in varicolored scenes (e.g., O-HAZE), dramatic performance drop can be observed, indicating its effectiveness for varicolored scenes.

Table 3: Ablation study on different components in $\mathcal{L}(\tilde{x}_0, y)$.

| Settings | w/o $\mathcal{L}_{sc}$ | | w/o $\mathcal{L}_{cc}$ | | w/o $\mathcal{L}_{wb}$ | | w/o $\mathcal{L}_{dcp}$ | | AECRNet-DIffAD | |
| --- | --- | --- | --- | --- | --- | --- | --- | --- | --- | --- |
| | PSNR↑ | SSIM↑ | PSNR↑ | SSIM↑ | PSNR↑ | SSIM↑ | PSNR↑ | SSIM↑ | PSNR↑ | SSIM↑ |
| OTS / SOTS-indoor | 24.46 | 0.8885 | 25.00 | 0.9123 | - | - | 22.98 | 0.9063 | **25.05** | **0.9224** |
| Wu / O-HAZE | 18.34 | 0.6815 | 19.07 | 0.7906 | 17.99 | 0.7684 | 18.66 | 0.7899 | **19.11** | **0.7911** |

## 5.3 COMPARISONS WITH REAL IMAGE DEHAZING METHODS

**Datasets and Evaluation Metrics.** To evaluate the real-world dehazing performance of the proposed DiffAD, we conduct experiments on real-world datasets, including labeled dataset O-HAZE (Ancuti et al., 2018a), I-HAZE (Ancuti et al., 2018b), NH-HAZE (Ancuti et al., 2020), and unlabeled dataset RTTS (Li et al., 2018). For labeled datasets, we adopt PSNR and SSIM as evaluation metrics. For unlabeled dataset, three non-reference image quality assessment (NRIQA) metrics, BRISQUE (Mittal et al., 2012), MUSIQ (Ke et al., 2021) and CLIPIQA (Wang et al., 2023) are utilized to evaluate the dehazing performance.

**Implementation Details.** We select the FocalNet (Cui et al., 2023) (pre-trained on Wu's dataset (Wu et al., 2023)) as our underlying model. Following (Shao et al., 2020; Chen et al., 2021), we utilize real-captured hazy images from URHI dataset (Li et al., 2018) and generate corresponding high-quality pseudo-labels via our DiffAD pipeline. We set $k = 50$ and $\lambda_{dcp} = 1e^{-3}$ in DiffAD and use the automatic method described in Sec. 4.2. We fine-tune FocalNet for 100 epochs with batch size set to 16 and learning rate set to $1e^{-4}$. We denote the fine-tuned model as DiffAD-FT, and fine-tune another light-weight model (DiffAD-S-FT) by removing depth estimation and SFT layers (Wang et al., 2018).

Table 4: Quantitative comparisons of various dehazing methods on real-captured hazy datasets.

| Method | O-HAZE | | I-HAZE | | NH-HAZE | | RTTS | | |
| --- | --- | --- | --- | --- | --- | --- | --- | --- | --- |
| | PSNR↑ | SSIM↑ | PSNR↑ | SSIM↑ | PSNR↑ | SSIM↑ | BRISQUE↓ | MUSIQ↑ | CLIPIQA↑ |
| (CVPR'20) DAD | 18.36 | 0.7484 | 18.02 | 0.7982 | 14.34 | 0.5564 | 32.37 | 49.88 | 0.2544 |
| (CVPR'21) PSD | 11.66 | 0.6831 | 13.79 | 0.7379 | 10.62 | 0.5246 | 21.62 | 52.81 | 0.2497 |
| (CVPR'22) D4 | 16.96 | 0.7229 | 15.64 | 0.7294 | 12.67 | 0.5043 | 32.21 | 53.57 | 0.3401 |
| (CVPR'23) RIDCP | 16.52 | 0.7154 | 16.88 | 0.7794 | 12.32 | 0.5341 | 17.29 | 59.38 | 0.3366 |
| (Ours) DiffAD-S-FT | **19.12** | **0.8072** | 18.14 | **0.8429** | 12.95 | **0.5661** | **15.41** | **60.38** | **0.3791** |
| (Ours) DiffAD-FT | **20.02** | **0.8155** | **18.59** | 0.8338 | **14.60** | **0.5805** | 14.73 | 60.18 | 0.3717 |

**Performance Evaluation.** We compare our DiffAD-FT with state-of-the-art real-world image dehazing methods: DAD (Shao et al., 2020), PSD (Chen et al., 2021), D4 (Yang et al., 2022) and RIDCP (Wu et al., 2023). We summarize the quantitative results of SOTA methods in Table 4. Our DiffAD-FT outperforms competing methods by a large margin on all of four datasets. Our

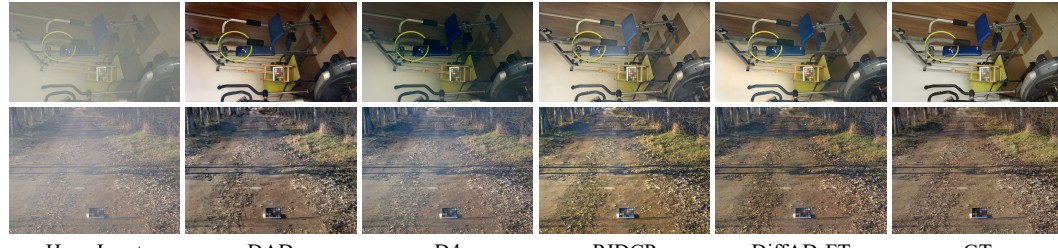

Hazy Input     DAD     D4     RIDCP     DiffAD-FT     GT

Figure 7: Dehazing results of various methods on labeled datasets. Top: I-HAZE. Bottom: O-HAZE.

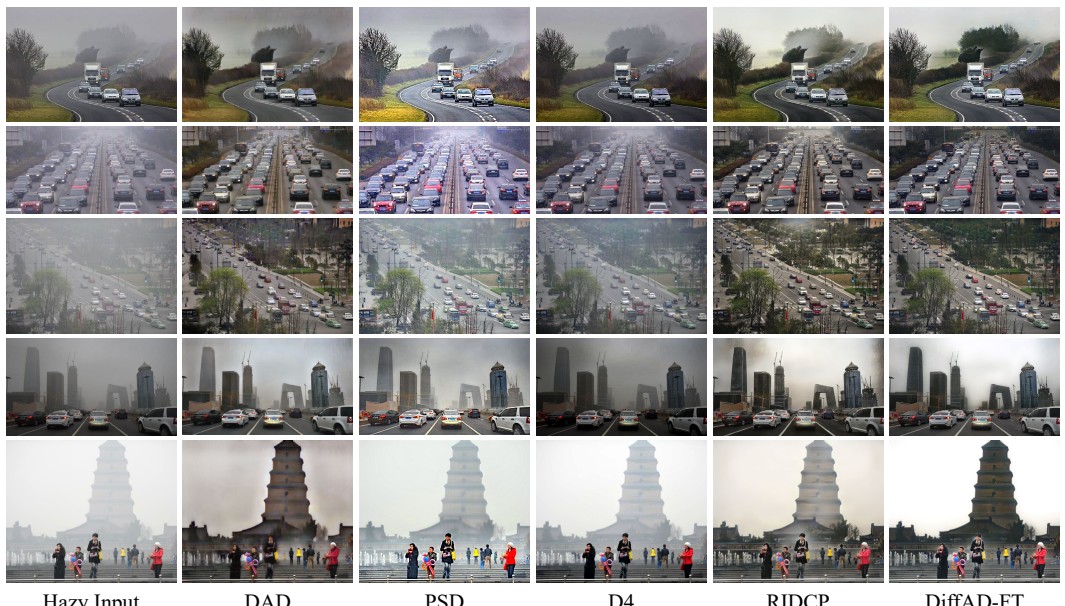

Hazy Input     DAD     PSD     D4     RIDCP     DiffAD-FT

Figure 8: Dehazing results of various methods on RTTS dataset.

DiffAD-S-FT also achieves promising performance. The qualitative results on labeled datasets and the unlabeled dataset are shown in Fig. 7 and Fig. 8, respectively. It can be observed that the results generated by our DiffAD-FT maintain higher visibility and fewer artifacts when compared with SOTA methods.

## 6 LIMITATION AND CONCLUSION

**Limitation.** By studying our DiffAD, we observe some difficulties that are urgent to be addressed. (1) We find it's hard to properly evaluate the dehazing performance by current metrics, especially in real image dehazing where the ground-truth is not available. (2) It is sub-optimal to fix hyper-parameters when generating pseudo labels. We plan to make them input-adaptive in future.

**Conclusion.** In this paper, we propose a novel **Diff**usion-based **AD**aptation paradigm (i.e., DiffAD) to explore the domain shift problem in image dehazing. We train a denoising diffusion probabilistic model (DDPM) with source hazy images to capture the prior probability distribution of the source domain. A source-Gaussian-source loop is built and given a hazy image from the target domain (e.g., real-captured hazy image), we can adjust the distribution to make it align with the source domain. Then, the adapted hazy image can be directly fed into a certain SOTA dehazing model pre-trained on source domain to predict the haze-free output. The proposed DiffAD can be successfully applied to real image dehazing by employing the predicted haze-free outputs as the pseudo labels.

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

# A   APPENDIX

## A.1   VISUAL RESULTS GENERATED BY DIFFUSION MODEL

To validate that the trained diffusion model can learn the haze distribution of the source domain, we utilize the diffusion model trained on ITS and OTS to generate 100 indoor and 100 outdoor hazy images, respectively. For comparison, we also randomly sample 100 indoor hazy images and outdoor hazy images from the two source domain (i.e., ITS dataset and OTS dataset), respectively. As shown in Fig. 9 (a)-(d), generated hazy images is similar to the original hazy images of the source domains.

Furthermore, we leverage VGG19 (Simonyan & Zisserman, 2014) to extract features from these 400 hazy images and apply t-SNE for dimensionality reduction, as shown in Fig. 9 (e). It can be observed that the generated source domain images are intertwined with the original source domain images on the t-SNE map, while hazy images from different source domains remain separated from one another. This further validate that our trained diffusion model can effectively capture the haze distribution of the source domain.

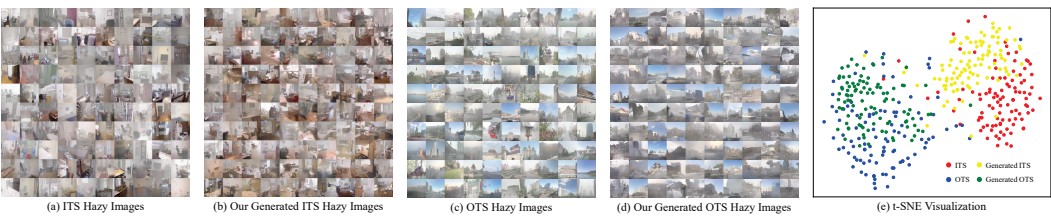

| (a) ITS Hazy Images | (b) Our Generated ITS Hazy Images | (c) OTS Hazy Images | (d) Our Generated OTS Hazy Images | (e) t-SNE Visualization |

Figure 9: Q3 of Reviewer-9KzP: Visualization of hazy images generated by diffusion models trained on different source domains.

## A.2   CONTROLLABILITY OF DIFFAD

In DiffAD, quality loss $\mathcal{Q}(\tilde{x}_0, y)$ allows user to control the reverse process from two perspectives, *i.e.*, color tone and dehazing effect. As shown in Fig. 10 and Fig. 11, users can adjust $\lambda_{wb}$ and $\lambda_{dcp}$ to achieve the desired output according to their preferences.

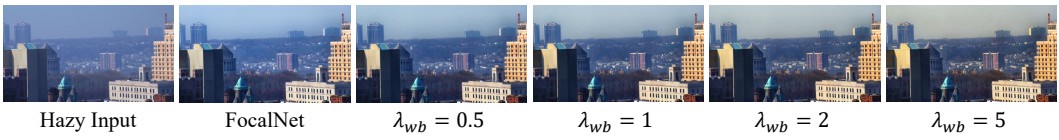

| Hazy Input | FocalNet | $\lambda_{wb} = 0.5$ | $\lambda_{wb} = 1$ | $\lambda_{wb} = 2$ | $\lambda_{wb} = 5$ |

Figure 10: Visual results of DiffAD with different $\lambda_{wb}$

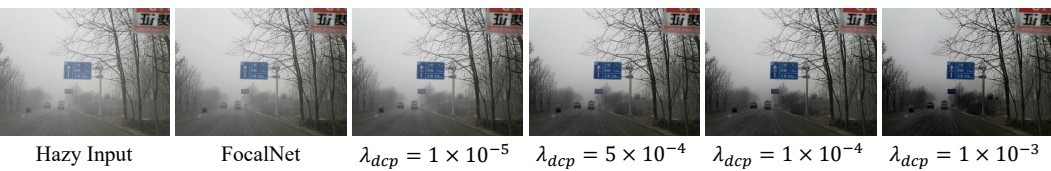

| Hazy Input | FocalNet | $\lambda_{dcp} = 1 \times 10^{-5}$ | $\lambda_{dcp} = 5 \times 10^{-4}$ | $\lambda_{dcp} = 1 \times 10^{-4}$ | $\lambda_{dcp} = 1 \times 10^{-3}$ |

Figure 11: Visual results of DiffAD with different $\lambda_{dcp}$

## A.3   ADDITIONAL VISUAL RESULTS ON SCENE TYPE ADAPTATION

We include some dehazing results in synthetic dense hazy scenes in Fig. 12. The qualitative results demonstrate that our DiffAD is also robust in dense hazy conditions.

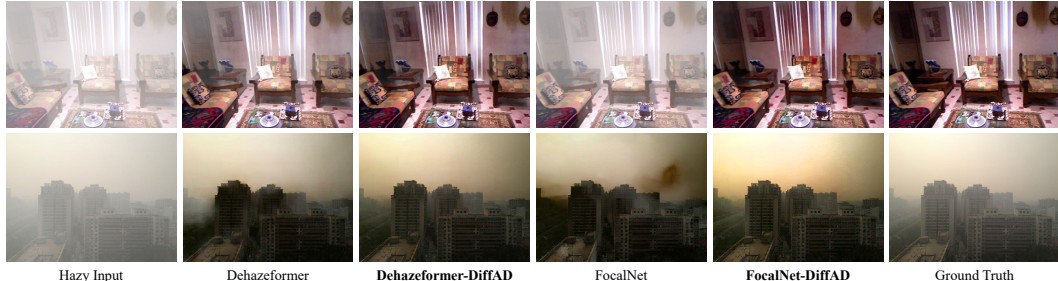

Figure 12: Q1 of Reviewer-yP7K: Visual results of scene type adaptation (dense hazy scenes are selected here). Top: qualitative results under "OTS / SOTS-indoor" setting. Bottom: qualitative results under "ITS / SOTS-outdoor" setting.

## A.4 ADDITIONAL VISUAL RESULTS ON HAZE TYPE ADAPTATION

We provide the visual results of haze type adaptation in Fig. 13. Significant dehazing performance improvement can be observed.

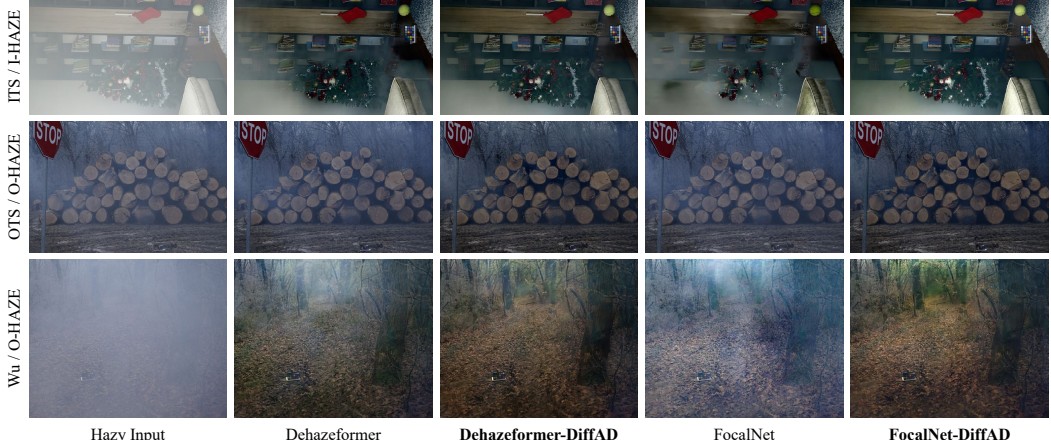

Figure 13: Q3 of Reviewer-fxe7: Visual results of haze type adaptation. Top: qualitative results under "ITS / I-HAZE" setting. Middle: qualitative results under "OTS / O-HAZE" setting. Bottom: qualitative results under "Wu / O-HAZE" setting.

We further adopt the real-world RTTS dataset (Li et al., 2018) to evaluate the effectiveness of our DiffAD. Specifically, we select models trained on ITS and OTS datasets to test their performance on RTTS dataset. The qualitative results are presented in Fig. 14.

## A.5 ADDITIONAL VISUAL RESULTS OF ABLATION STUDY

We provide additional visual ablation study in Fig. 15. Removing the spatial consistency loss $\mathcal{L}_{sc}$ (Fig. 15 (b)) introduces many artifacts in the dehazing results due to the generative nature of the diffusion model, thus failing the preservation of structure information. Discarding the color consistency loss $\mathcal{L}_{cc}$ (Fig. 15 (c)) hinders the preservation of original vivid local color information. This is because the diffusion model, in the reverse process, alters not only the structural information but also the local color information. As shown in Fig. 15 (d), the results without white balance loss $\mathcal{L}_{wb}$ exhibit severe color casts when encountering varicolored hazy scenes. When the region-aware DCP loss $\mathcal{L}_{rdcp}$ is absence, more haze residue in the dehazing reuslts, as indicated by Fig. 15 (e).

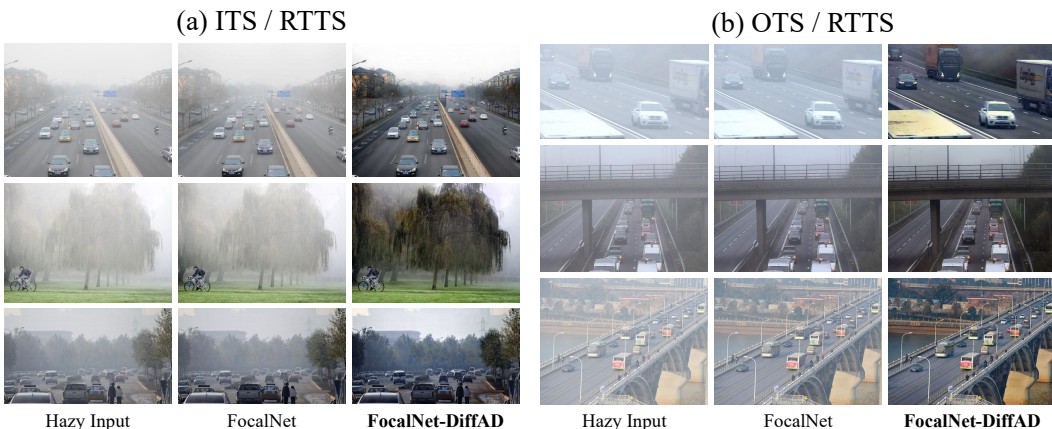

(a) ITS / RTTS        (b) OTS / RTTS

Hazy Input    FocalNet    **FocalNet-DiffAD**     Hazy Input    FocalNet    **FocalNet-DiffAD**

Figure 14: Q2 of Reviewer-V9fx and Q2 of Reviewer-fxe7: Visual results of haze type adaptation. Left: qualitative results under "ITS / RTTS" setting. Right: qualitative results under "OTS / RTTS" setting.

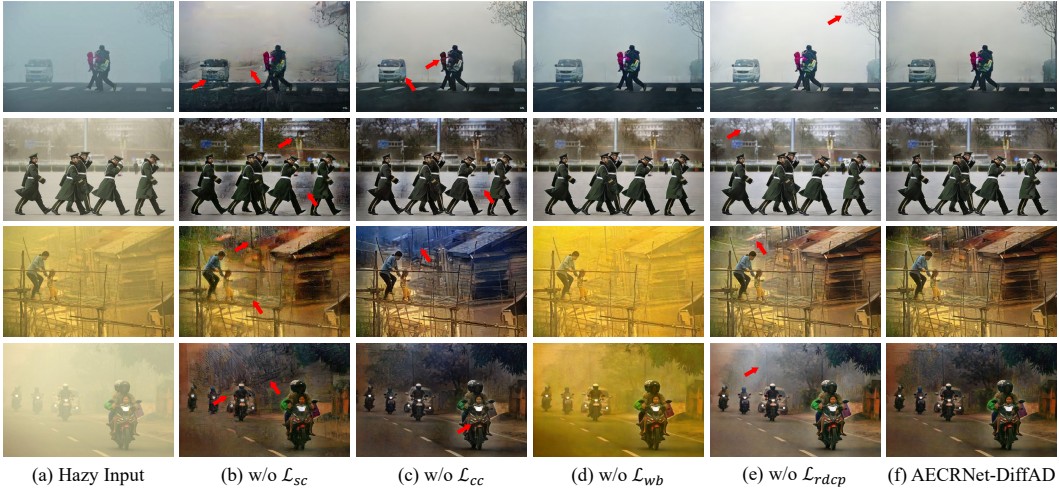

(a) Hazy Input    (b) w/o $\mathcal{L}_{sc}$    (c) w/o $\mathcal{L}_{cc}$    (d) w/o $\mathcal{L}_{wb}$    (e) w/o $\mathcal{L}_{rdcp}$    (f) AECRNet-DiffAD

Figure 15: Q2 of Reviewer-yP7K and Q2 of Reviewer-9KzP: Ablation study of each loss on RTTS dataset.

## A.6   DETAILED ARCHITECTURE OF SFT LAYER

Considering haze is highly related to the scene depth, we embed the depth imformation into the encoder of the dehazing network to guide the dehazing process. Specifically, for hazy features $F_{hazy}$ extracted in each level of the dehazing encoder, we first utilize convolution layer to extract the depth features $F_{depth}$ with the same dimensions from the estimated depth map. Then, we utilize SFT layer (Wang et al., 2018) to achieve effective modulation of $F_{hazy}$ and $F_{depth}$. The structure of SFT layer is illustrated in Fig. 16. Two groups of different convolution layers are adopted to predict scale parameter $\gamma$ and shift parameter $\beta$. Transforming the hazy features $F_{hazy}$ with predicted parameters, we can obtain the modulated features $F_{out}$:

$$F_{out} = SFT(F_{hazy}|\gamma, \beta) = (1 + \gamma) \cdot F_{hazy} + \beta \tag{17}$$

## A.7   ABLATION STUDY OF DIFFAD-FT

We conduct ablation study to verify the effectiveness of each component, *i.e.*, embedding depth and refining pseudo labels, of the fine-tune process.

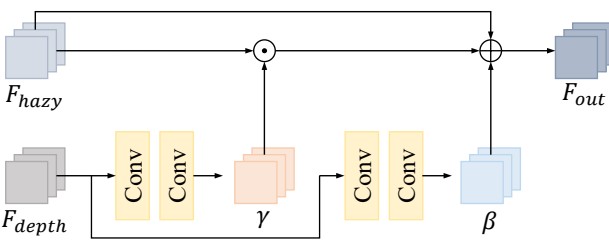

Figure 16: Structure of SFT layer.

The qualitative results on three NTIRE datasets (Ancuti et al., 2018a;b; 2020) are summarized in Table 5. It can be observed that both embedding depth information and refining pseudo label play important roles in fine-tune process. Furthermore, we presents visual results in Fig. 17. On the one hand, without depth information, the fine-tuned model struggles to model haze density perfectly, resulting in residual haze in the dehazed result (Fig. 17 (b)). On the other hand, skipping refinement when generating pseudo labels leads to over-enhancement in sky regions (Fig. 17 (c)).

Table 5: Ablation study of DiffAD-FT on NTIRE datasets (Ancuti et al., 2018a;b; 2020)

| Settings | O-HAZE | | I-HAZE | | NH-HAZE | |
|---|---|---|---|---|---|---|
| | PSNR↑ | SSIM↑ | PSNR↑ | SSIM↑ | PSNR↑ | SSIM↑ |
| w/o Depth | 19.12 | 0.8072 | 18.14 | **0.8429** | 12.95 | 0.5661 |
| w/o Refine | 19.68 | 0.8134 | 15.65 | 0.7888 | 13.89 | 0.5651 |
| DiffAD-FT | **20.02** | **0.8155** | **18.59** | 0.8338 | **14.60** | **0.5805** |

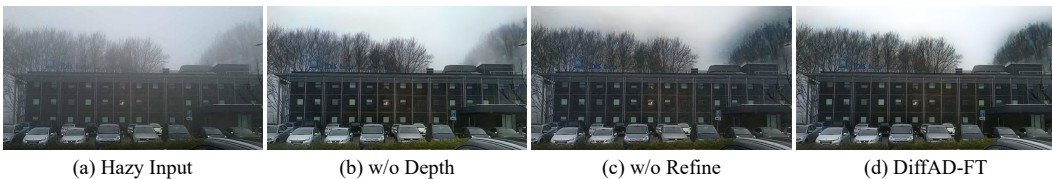

| (a) Hazy Input | (b) w/o Depth | (c) w/o Refine | (d) DiffAD-FT |
|---|---|---|---|

Figure 17: Visual results of the ablation study on DiffAD-FT

## A.8 EFFICIENCY ANALYSIS OF DIFFAD-FT

DiffAD can effectively reduce the domain shift, but struggles to be applied in real-world due to limited efficiency. To this end, we propose DiffAD-FT and its light-weight version DffAD-S-FT to improve the efficiency while maintaining good generalization ability. The runtime comparisons between fine-tuned models and DiffAD are summarized in Table 6. Compared to DiffAD, DiffAD-FT and DiffAD-S-FT achieve significant improvement in runtime.

Table 6: Runtime comparisons. The runtime is measured on $512 \times 512$ images using a single NVIDIA RTX 3090 GPU.

| Methods | DiffAD | | DiffAD-FT | DiffAD-S-FT |
|---|---|---|---|---|
| | $k = 10$ | $k = 50$ | | |
| Runtime (ms) | 2344 | 11147 | 121.57 | 17.82 |

## A.9 ADDITIONAL VISUAL RESULTS

More visual results on RTTS (Li et al., 2018) are shown in Fig. 18 and Fig. 19. We also provide some visual results on Fattal's dataset (Fattal, 2014) in Fig. 20. We can observe that our DiffAD-FT achieves more visual pleasing results in terms of less artifacts and haze residue when competing with other SOTA methods. We also provide results of our DiffAD-FT on dense hazy scenes in Fig. 21.

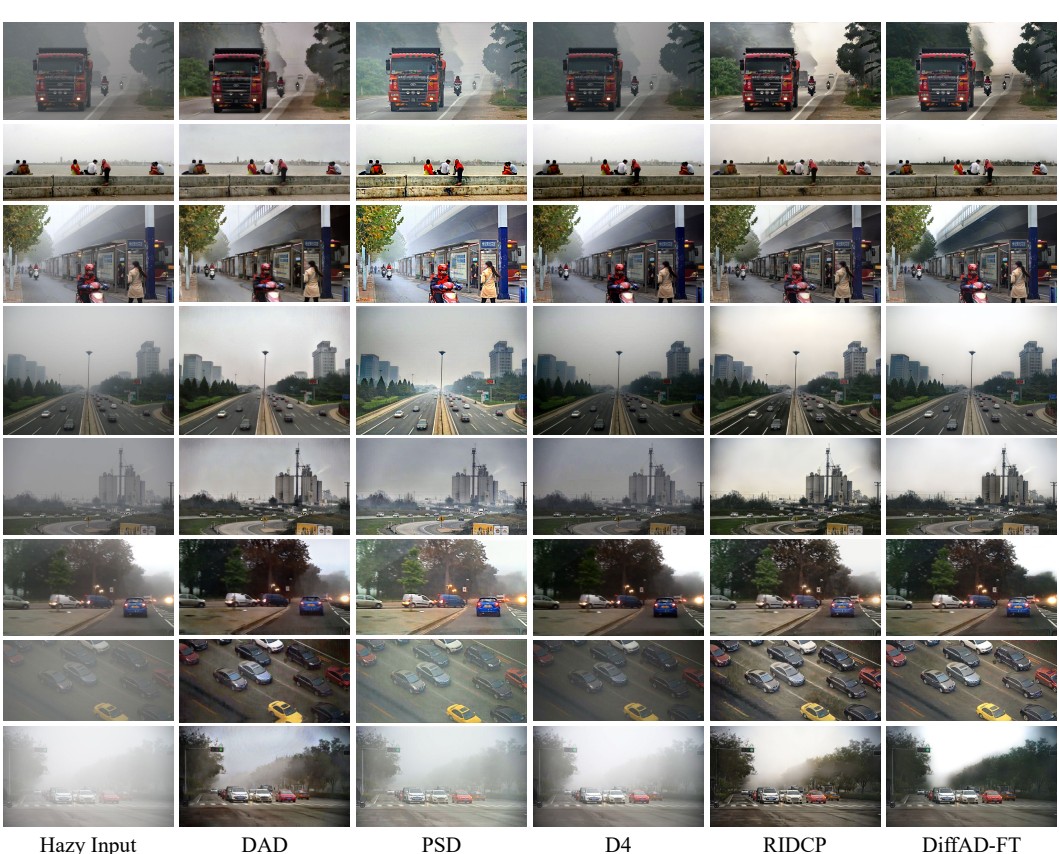

Hazy Input          DAD          PSD          D4          RIDCP          DiffAD-FT

Figure 18: Dehazing results of various methods on RTTS dataset. Please zoom in on screen for a better view.

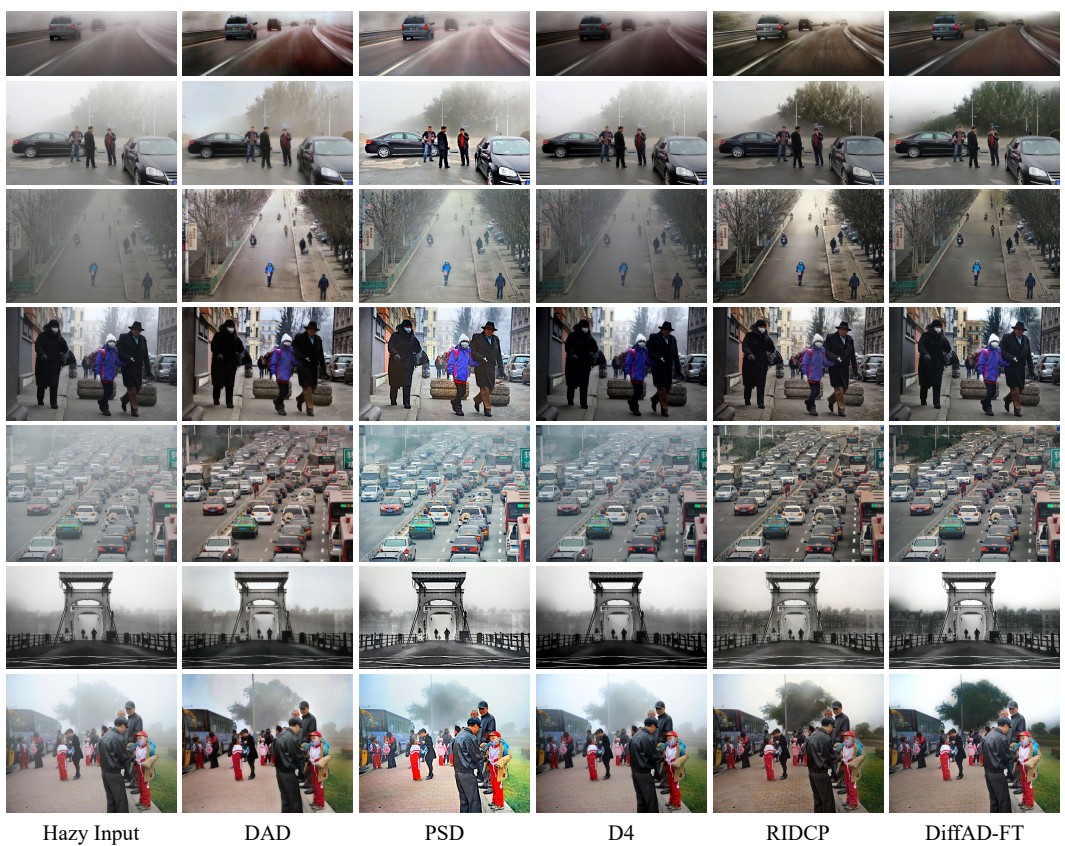

Hazy Input          DAD          PSD          D4          RIDCP          DiffAD-FT

Figure 19: Dehazing results of various methods on RTTS dataset. Please zoom in on screen for a better view.

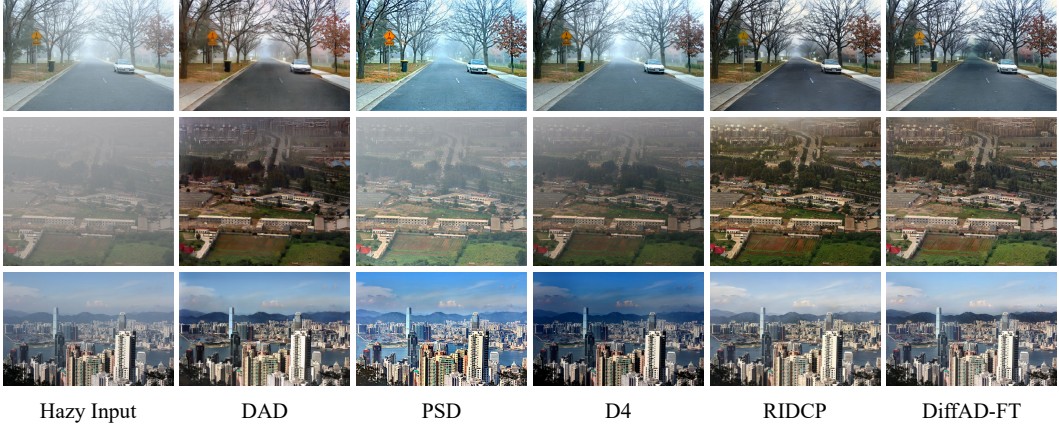

Hazy Input          DAD          PSD          D4          RIDCP          DiffAD-FT

Figure 20: Dehazing results of various methods on Fattal's dataset. Please zoom in on screen for a better view.

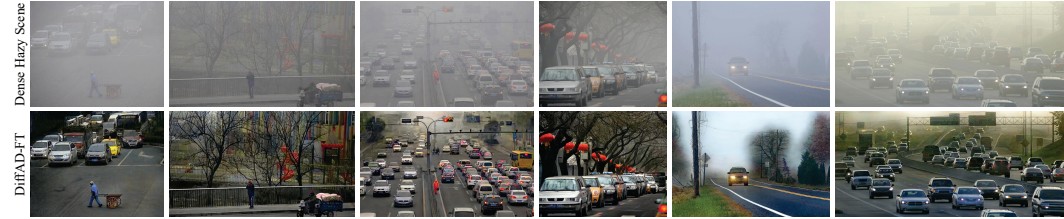

Figure 21: Q1 of Reviewer-yP7K: Visual results of our DiffAD-FT on dense hazy scenes.

