# OpenReview forum: "Addressing domain shift with diffusion-based adaptation for real image dehazing"
_ICLR.cc/2025/Conference — ICLR 2025 Conference Withdrawn Submission_

### Official Review · Reviewer_fxe7 · 2024-11-01

**Soundness:** 3
**Presentation:** 3
**Contribution:** 3
**Rating:** 5
**Confidence:** 5

**Summary:**

This paper proposes a Diffusion-based Adaptation pipeline to solve the domain shift problem in image dehazing. The main contribution of this paper is to use conditional generation to ensure that no structural deformation and color distortion are introduced in the generated images. Another contribution is the use of multiple losses to ensure the fidelity and quality of the generated images.

**Strengths:**

This paper is well organized. The topic of the paper is very good. How to use synthetic data to improve the performance on real-world hazy images should be the research trend in the field of image dehazing in the future.

**Weaknesses:**

1. Using models trained on indoor datasets to handle outdoor scenes does not seem to be the focus of this paper. I think the biggest use of DiffAD is to improve the performance of models trained on synthetic hazy-clean image pairs when used to process real-world hazy images.
2. The problem with the dataset selected for the Haze Type Adaptation (between synthetic and real) experiment. The I-Haze and O-Haze datasets are haze generated using a professional haze generator and are not real-world hazy images in nature.
3. As shown in Table 2, the PSNR after using DiffAD is improved, but still very low. How does the dehazed result look like? This paper lacks qualitative results of this experiment.

**Questions:**

Please refer to Weaknesses.

---

> ### Author Response · Authors · 2024-11-24
> **Author Response 1 (highlighted with red in revised PDF)**
>
> **Q1**: Using models trained on indoor datasets to handle outdoor scenes does not seem to be the focus of this paper. I think the biggest use of DiffAD is to improve the performance of models trained on synthetic hazy-clean image pairs when used to process real-world hazy images.
>
> **A1**: Thank you for pointing this out. The initial goal of scene type adaptation is to validate the capability of DiffAD in adapting input images from out-of-distribution back to source domain. Subsequently, the dehazing model trained on the source domain can effectively process such input images.
>
> As you have stated (we can’t agree more), the ultimate objective of DiffAD is to apply the models trained on synthetic datasets to tackle real-world image dehazing task. It is worth mentioning that our topic (Title: Exploring domain shift with diffusion-based adaptation for real image dehazing) is specifically focused on real image dehazing.
>
> ---
>
> **Q2**: The problem with the dataset selected for the Haze Type Adaptation (between synthetic and real) experiment. The I-Haze and O-Haze datasets are haze generated using a professional haze generator and are not real-world hazy images in nature.
>
> **A2**: According to this insightful comment, we further perform experiments on real-world nature hazy images. Specifically, we choose the model pre-trained on ITS/OTS (synthetic) to test the performance on RTTS (real).
>
> Since there is no aligned clean images, we employ some no-reference image quality assessment (NRIQA) metrics to quantitatively evaluate the performance. The quantitative results are listed in the following table.
>
> | Settings                   | ITS/RTTS | ITS/RTTS  | OTS/RTTS | OTS/RTTS  |
> |----------------------------|----------|-----------|----------|-----------|
> |                            | BRISQUE$\downarrow$  | PaQ-2-PiQ$\uparrow$ | BRISQUE$\downarrow$  | PaQ-2-PiQ$\uparrow$|
> | (CVPR'21) AECRNet          | 29.40    | 66.30     | 30.64    | 66.56     |
> | (Ours) AECRNet-DiffAD      | **28.43**    | **67.69**     | **28.83**    | **66.79**     |
> | (TIP'23) Dehazeformer      | 30.17    | 66.61     | 33.16    | 66.60     |
> | (Ours) Dehazeformer-DiffAD | **29.49**    | **67.38**     | **30.10**    | **66.78**     |
> | (ICCV'23) FocalNet         | 33.75    | 66.35     | 35.82    | 66.48     |
> | (Ours) FocalNet-DiffAD     | **31.53**    | **67.48**     | **32.15**    | **66.93**     |
>
> We can observe that the selected models with DiffAD can achieve robust improvements on RTTS.
>
> In addition, we also provide some qualitative results in Fig. 14 (**please refer to our revised PDF**).
>
> ---
>
> **Q3**: As shown in Table 2, the PSNR after using DiffAD is improved, but still very low. How does the dehazed result look like? This paper lacks qualitative results of this experiment.
>
> **A3**: We provide the qualitative results corresponding to Table 2, please refer to Fig. 13 in the appendix. We can observe from Fig. 13 that after utilizing DiffAD, the learned dehazing priors are activated, resulting in a significant improvement in dehazing performance.

---

> > ### Author Response · Authors · 2024-11-30
> >
> > Dear Reviewer fxe7,
> >
> > We would like to express our gratitude for your valuable feedback.
> > We have carefully considered all suggestions and updated our paper accordingly.
> >
> > At present, we are eager to know if we have adequately addressed your questions and concerns.
> > Kindly hope you can help us again in further improving this work.
> >
> > Thank you for your hard work and support.
> >
> > Best,
> >
> > Authors

---

### Official Review · Reviewer_tedF · 2024-11-01

**Soundness:** 3
**Presentation:** 2
**Contribution:** 2
**Rating:** 6
**Confidence:** 3

**Summary:**

The paper presents a diffusion-based domain adaptation method aimed at enhancing image dehazing. It utilizes a pre-trained denoising diffusion probabilistic model to transform hazy images from the target domain into the source domain, where existing dehazing methods are trained and thus demonstrate superior performance. This approach not only improves the performance of previous dehazing techniques in real-world scenarios but also outperforms other real-world dehazing methods across various datasets.

**Strengths:**

The method demonstrates several strengths:

1. This approach effectively enhances the performance of previous dehazing techniques in both synthetic and real-world scenarios.

2. The work surpasses other real-world dehazing methods across various datasets.

3. The concept of diffusion-based domain adaptation for dehazing is novel.

**Weaknesses:**

1. The citation format in the manuscript should be revised. For example, "ill-posed problem Wu et al. (2021); Chen et al. (2024a)" should be formatted as "ill-posed problem (Wu et al., 2021; Chen et al., 2024a)."

2. The contributions of the work may be considered minor. The research builds on existing approaches, including controllable diffusion processes, DCP loss, and spatial consistency loss. The direct application and combination of these works suggest a lack of significant contributions. However, in my personal opinion, the idea of utilizing diffusion-based domain adaptation to enhance dehazing is both intuitive and impressive.

3. The direct masked addition between two dehazed images in Figure 5 may lead to inconsistencies. Additionally, the sky and depth masks operate within different value ranges, making their direct addition insufficient in detail.

4. The manuscript presents only a limited number of visual comparisons. It would be beneficial to include more visual results in the blank space on Page 10, potentially with additional figures in the appendix.

5. The title "ADDRESSING DOMAIN SHIFT" may be considered an overstatement, as several unresolved problems remain, and not all cases have been evaluated and compared.

The work presents an interesting idea and motivation; however, the current manuscript would benefit from further refinement. The contributions appear to be around the borderline, but I personally recommend leaning towards acceptance.

**Questions:**

N/A

---

> ### Author Response · Authors · 2024-11-24
> **Author Response 1 (highlighted with orange in revised PDF)**
>
> **Q1**: The citation format in the manuscript should be revised. For example, "ill-posed problem Wu et al. (2021); Chen et al. (2024a)" should be formatted as "ill-posed problem (Wu et al., 2021; Chen et al., 2024a).”
>
> **A1**: We appreciate your suggestion and have checked the TeX file meticulously to ensure that the compiled output meets the required standards. Specifically, `\cite{}` is replaced with `\citet{}` (when the authors are included in the sentence) or `\citep{}` (otherwise).
>
> ---
>
> **Q2**: The contributions of the work may be considered minor. The research builds on existing approaches, including controllable diffusion processes, DCP loss, and spatial consistency loss. The direct application and combination of these works suggest a lack of significant contributions. However, in my personal opinion, the idea of utilizing diffusion-based domain adaptation to enhance dehazing is both intuitive and impressive.
>
> **A2**: We want to thank this reviewer for acknowledging the merits of our idea.
>
> Our DiffAD is the first method to utilize the diffusion model for domain projection in image dehazing. To the best of our knowledge, this is the first time that the diffusion model has been employed to transfer the probability distribution of real-world hazy images (i.e., input) into the synthetic domain (i.e., synthetic hazy). The adapted hazy input is then fed into a pre-trained dehazing model to remove the haze. Consequently, the learned dehazing priors from the synthetic domain can be effectively utilized. In our DiffAD framework, the diffusion bridge serves as a preprocessing step prior to the pre-trained dehazing model. DiffAD provides a novel scheme for real image dehazing and has achieved promising results.
>
> We re-organize the contribution part (please check the revised PDF).
>
> In summary, our main contributions are as follows:
>
> - We propose a novel **Diff**usion-based **AD**aptation paradigm (i.e., DiffAD) to explore the domain shift problem in image dehazing. To the best of our knowledge, this is the first time that the diffusion model has been employed to transfer the probability distribution of target domain (e.g., real-world hazy) into the source domain (e.g., synthetic hazy). DiffAD is a plug-and-play module that acts on the input image, thus will not alter the underlying dehazing model. The dehazing priors encapsulated in the underlying dehazing model can be fully explored and exploited.
> - To guide the generation during the reverse process, a novel loss function is devised from the perspective of fidelity and quality. We show that the fidelity item can avoid information loss and the quality item brings controllability, ensuring the generation of high-quality haze-free images.
> - We further take the obtained haze-free images as the pseudo labels to fine-tune the underlying dehazing model. This updated model can be directly applied to recover real-world hazy images with enhanced efficiency.
>
> In addition, we re-write Sec. 4.1.2 to explain the motivations behind the loss functions. Please refer to **Reviewer 9KzP’s A2** for more information.
>
> ---
>
> **Q3**: The direct masked addition between two dehazed images in Figure 5 may lead to inconsistencies. Additionally, the sky and depth masks operate within different value ranges, making their direct addition insufficient in detail.
>
> **A3**: We want to thank this reviewer for reminding us about the potential value inconsistency issue.
>
> Since the depth map $\mathcal{W}\_{D}$ obtained through depth estimation represents relative depth, the values range between [0, 1]. Similarly, the output of the sky estimation is a nearly binary sky mask $\mathcal{W}\_{S}$ with values also ranging between [0, 1], primarily either 0 or 1. Therefore, both depth map $\mathcal{W}\_{D}$ and sky mask $\mathcal{W}\_{S}$ share the same value range.
>
> As shown in Fig. 5, in $\mathcal{W}\_{D}$, sky regions are typically represented as 0, whereas in $\mathcal{W}\_{S}$, sky regions are generally represented as 1, and other regions are close to 0. Thus, directly adding $\mathcal{W}\_{D}$ and $\mathcal{W}\_{S}$ is equivalent to performing an inversion operation in the sky region of the depth map $\mathcal{W}\_{D}$. Additionally, after obtaining the weight map $\mathcal{W}$ by adding $\mathcal{W}\_{D}$ and $\mathcal{W}\_{S}$, we normalize the $\mathcal{W}$ to [0, 1].
>
> In conclusion, this approach is simple and efficient, and faces a very small risk of introducing inconsistencies.
>
> ---
>
> **Q4**: The manuscript presents only a limited number of visual comparisons. It would be beneficial to include more visual results in the blank space on Page 10, potentially with additional figures in the appendix.
>
> **A4**: According to this helpful comment, we have added more visual comparisons in Page 10. Please check our revised PDF.

---

> ### Author Response · Authors · 2024-11-24
> **Author Response 2 (highlighted with orange in revised PDF)**
>
> **Q5**: The title "ADDRESSING DOMAIN SHIFT" may be considered an overstatement, as several unresolved problems remain, and not all cases have been evaluated and compared.
>
> **A5**: Thank you for pointing this out for us. To avoid overstatement, we plan to revise our title to “Exploring Domain Shift with Diffusion-based Adaptation for Real Image Dehazing”.

---

> > ### Author Response · Authors · 2024-11-30
> >
> > Dear Reviewer tedF,
> >
> > We would like to express our gratitude for your valuable feedback.
> > We have carefully considered all suggestions and updated our paper accordingly.
> >
> > At present, we are eager to know if we have adequately addressed your questions and concerns.
> > Kindly hope you can help us again in further improving this work.
> >
> > Thank you for your hard work and support.
> >
> > Best,
> >
> > Authors

---

### Official Review · Reviewer_yP7K · 2024-11-02

**Soundness:** 2
**Presentation:** 3
**Contribution:** 2
**Rating:** 6
**Confidence:** 5

**Summary:**

This paper introduces Diffusion-based Adaptation (DiffAD), a method designed to address out-of-distribution haze images without altering the architecture or weights of traditional dehazing models. DiffAD leverages a denoising diffusion model (DDPM) to transform the distribution of input images from the target domain to the source domain. Specifically, the DDPM is trained on source haze images, and during test-time, target haze inputs are adapted to resemble source images using the reverse diffusion process. Additionally, the method refines the model using pseudo labels derived from its predictions.

**Strengths:**

1. The motivation of addressing the domain gaps is meaningful.
2. The structure of the paper is clear and well-organized.
3. The proposed method achieves state-of-the-art performance on multiple datasets.

**Weaknesses:**

1. The experimental results are not particularly impressive. For example, the provided qualitative results only demonstrate sparse fog.

2. Some ablation studies are missing. Specifically, the effectiveness of the fidelity loss and quality loss should be discussed.

3. The proposed framework seems idealized. However, depth estimation and pseudo labels may be inaccurate. How does the proposed method address real-world fog when depth estimation and pseudo labels are unreliable?

**Questions:**

1. It would be beneficial to include ablation studies to evaluate the impact of the fidelity loss and quality loss.
2. In Section 4.2, depth estimation is applied to both the predicted images (J*) and real hazy images. However, the performance may degrade if the input images are not fully clear. How do you ensure the robustness of this approach?
3. In Section 4.2 (lines 349-354), why is the artifact-free pseudo label obtained using output J*? If the dehazing model is diffusion-based, there is a risk of artifacts appearing in J*. Could you clarify this?
4. The provided qualitative results seem to lack cases with dense fog. Could you include examples with denser fog conditions, both synthetic and real-world?
5. Did you use the same parameters for both synthetic and real-world evaluations?

---

> ### Author Response · Authors · 2024-11-24
> **Author Response 1 (highlighted with brown in revised PDF)**
>
> **Q1**: The experimental results are not particularly impressive. For example, the provided qualitative results only demonstrate sparse fog. The provided qualitative results seem to lack cases with dense fog. Could you include examples with denser fog conditions, both synthetic and real-world?
>
> **A1**: We include some dehazing results in synthetic dense hazy scenes in Fig. 12. We also include some dehazing results in real dense hazy scenes in Fig. 21. The visual results demonstrate that our method achieves robust dehazing performance, even in dense hazy conditions.
>
> ---
>
> **Q2**: Some ablation studies are missing. Specifically, the effectiveness of the fidelity loss and quality loss should be discussed. It would be beneficial to include ablation studies to evaluate the impact of the fidelity loss and quality loss.
>
> **A2**: The effectiveness of each loss function is discussed in our manuscript, please refer to the Sec 5.2.  In this part, we conduct a more detailed analysis of the contribution of each loss function in fidelity loss (i.e., spatial consistency loss $\mathcal{L}\_{sc}$ and color consistency loss $\mathcal{L}\_{cc}$) and quality loss (i.e., white balance loss $\mathcal{L}\_{wb}$ and region-aware DCP loss $\mathcal{L}\_{rdcp}$). The quantitative results are shown in Table 3.
>
> We also provide additional visual ablation studies in Fig. 15. On the one hand, we use the fidelity loss to preserve the structure and color information of the input.  Removing the spatial consistency loss $\mathcal{L}\_{sc}$ (Fig. 15 (b)) introduces many artifacts in the dehazing results due to the generative nature of the diffusion model, thus failing the preservation of structure information. Discarding the color consistency loss $\mathcal{L}\_{cc}$ (Fig. 15 (c)) hinders the preservation of original vivid local color information. This is because the diffusion model, in the reverse process, alters not only the structural information but also the local color information. On the other hand, the quality loss is designed to improve the overall dehazing effect.  As shown in Fig. 15 (d), the results without white balance loss $\mathcal{L}\_{wb}$ exhibit severe color casts when encountering varicolored hazy scenes. When the region-aware DCP loss $\mathcal{L}\_{rdcp}$ is absence, more haze residue in the dehazing results, as indicated by Fig. 15 (e).
>
> ---
>
> **Q3**: The proposed framework seems idealized. However, depth estimation and pseudo labels may be inaccurate. How does the proposed method address real-world fog when depth estimation and pseudo labels are unreliable? In Section 4.2, depth estimation is applied to both the predicted images (J*) and real hazy images. However, the performance may degrade if the input images are not fully clear. How do you ensure the robustness of this approach?
>
> **A3**: We sincerely thank this reviewer for giving us this wonderful inspiration. To mitigate the potential inaccurate predictions of pseudo labels, we have reviewed many related papers and conduct an experiment with confidence map. To be more specific, following [1], we estimate the confidence map $c$ from image features and dehazing results. We use the confidence map $c$ to mitigate inaccurate predictions by modifying the $\mathcal{L}\_1$ loss to $\mathcal{L}\_1 = ||c \cdot \Phi(I) -c \cdot J||\_1$, where $I$ denotes the hazy input and $J$ denotes the corresponding haze-free ground truth. $\Phi$ is the dehazing network.
>
> To avoid the confidence map $c$ becomes a full-zero vector when minimizing the $\mathcal{L}\_1$ loss, we supervise the confidence map with the loss *$\mathcal{L}\_c=-\frac{1}{N\times H\times W}\sum\_{n=1}^{N}\sum\_{j=1}^H\sum\_{k=1}^{W}{\log c\_{njk}}$*, where $N$, $H$, $W$ denote image channel, image height and image width respectively, and $c$ denotes the confidence map.
>
> The quantitative results are shown in the following table, w/ CM denotes introducing confidence map. Incorporating the confidence map further improves the performance of DiffAD-S-FT.
>
> |      Method       | PSNR on O-HAZE | SSIM on O-HAZE | PSNR on I-HAZE | SSIM on I-HAZE | PNSR on NH-HAZE | SSIM on NH-HAZE |
> | :---------------: | :------------: | :------------: | :------------: | :------------: | :-------------: | :-------------: |
> |    DiffAD-S-FT    |     19.12      |     0.8072     |     18.14      |   **0.8429**   |      12.95      |     0.5661      |
> | DiffAD-S-FT w/ CM |   **19.71**    |   **0.8132**   |   **18.27**    |     0.8407     |    **14.10**    |   **0.5789**    |
>
> [1] Rajeev, Yasarla, et al. “Uncertainty Guided Multi-Scale Residual Learning-using a Cycle Spinning CNN for Single Image De-Raining.” CVPR 2019.

---

> ### Author Response · Authors · 2024-11-24
> **Author Response 2 (highlighted with brown in revised PDF)**
>
> **Q4**: In Section 4.2 (lines 349-354), why is the artifact-free pseudo label obtained using output J*? If the dehazing model is diffusion-based, there is a risk of artifacts appearing in J*. Could you clarify this?
>
> **A4**: When generating pseudo-labels, thanks to the controllability of our pipeline, we can adjust the weight of the region-aware DCP loss $\lambda_{dcp}$ to control the residual haze in the results. However, manually adjusting an optimal  $\lambda_{dcp}$ for each image is obviously impractical. Therefore, to obtain haze-free pseudo-labels, we uniformly set the  $\lambda_{dcp}$  to a relatively large value (i.e., 0.001). However, for inputs with low haze density, large  $\lambda_{dcp}$ may lead to over-dehazing in sky regions, as shown in Fig. 5. Considering that the original output $J^*$ of the pre-trained dehazing model, while limited in dehazing performance, maintains good visual quality in sky regions (since these regions don't require much dehazing), we refine the DiffAD output $\tilde{J}$ with the pre-trained dehazing model’s output $J^*$ to generate the final pseudo-labels.
>
> In our implementation, to reduce computational overhead, we do not use a diffusion-based pre-trained dehazing network but instead employ a simple and efficient dehazing model, i.e., FocalNet, to generate pseudo-labels. Since FocalNet is a CNN-based network, its outputs inherently contain minimal artifacts, making it well-suited for correcting over-dehazing caused by the large $\lambda_{dcp}$.
>
> Moreover, even if $J^*$ itself may have artifacts, as noted in Q3, we find that introducing a confidence map can mitigate the negative impact of the inaccurate estimation of pseudo-labels.
>
> ---
>
> **Q5**: Did you use the same parameters for both synthetic and real-world evaluations?
>
> **A5**: For experiments of Table 1 and Table 2, we set $k = 10$, $\lambda_{sc} = 1$, $\lambda_{cc} = 0.1$, $\lambda_{wb} = 1$, and $\lambda_{dcp}=5e^{-5}$.
>
> For experiments of Table 4, we set $k = 50$, and $\lambda_{dcp}=1e^{-3}$. Here, we adopt relatively larger $k$ and $\lambda_{dcp}$ for the generation of high quality pseudo labels.

---

> > ### Comment · Reviewer_yP7K · 2024-11-26
> >
> > Dear authors,
> >
> > Thank you for your detailed response. I will maintain my score.
> >
> > Best regards,

---

> > > ### Author Response · Authors · 2024-11-27
> > >
> > > Dear Reviewer yP7K,
> > >
> > > We appreciate your great effort in reviewing our paper and thank you for checking our rebuttal.
> > >
> > > Best,
> > >
> > > Authors

---

### Official Review · Reviewer_9KzP · 2024-11-02

**Soundness:** 3
**Presentation:** 3
**Contribution:** 2
**Rating:** 5
**Confidence:** 5

**Summary:**

This paper proposes to tackle out-of-distribution hazy images through projecting to the source domain with diffusion model trained on source data, and then post-processed with existing dehazing model. Additionally, various loss functions are crafted for fidelity and quality. Experiments show the effectiveness of the proposed method.

**Strengths:**

1. This paper proposes to handle out-of-distribution hazy image with diffusion bridge to source domain.
2. Various loss functions are crafted for fidelity and quality.
3. Experiments show the Effectiveness of the proposed method.

**Weaknesses:**

1. Addressing the domain shift with diffusion bridge has been widely explored in literature, such as [DDBM, ICLR2024; Diffusion Schrödinger Bridge, NeurIPS2023], therefore, the contribution of this paper that projecting the out-of-distribution hazy images to source domain is somewhat less.
2. The loss functions are heavily handcrafted and yield lots of hyper-parameters, and basically build upon existing loss functions, such as grey-world-assumption loss, dcp loss, etc., which is somewhat less informative.
3. It is recommended to show the captured source distribution in the trained diffusion model for verification. Additionally, according to sec. 5.1, the diffusion model is only trained for 50k iterations from scratch, is this enough for diffusion model to learn the source distribution. Some generated source domain hazy image should be presented and compared with original source image.

**Questions:**

1. It is unclear when to add the proposed loss functions, are there added in test time as existing zero-shot inverse problem solver, or added in the training time.

---

> ### Author Response · Authors · 2024-11-24
> **Author Response 1 (highlighted with blue in revised PDF)**
>
> **Q1**: Addressing the domain shift with diffusion bridge has been widely explored in literature, such as [DDBM, ICLR2024; Diffusion Schrödinger Bridge, NeurIPS2023], therefore, the contribution of this paper that projecting the out-of-distribution hazy images to source domain is somewhat less.
>
> **A1**: We have carefully reviewed the papers recommended by the reviewers. The diffusion bridge and Schrödinger bridge in previous literature interpolate between two paired distributions given as endpoints. A typical application is image-to-image translation (e.g., from edge maps to colored handbags).
>
> Nevertheless, our DiffAD is the first method to utilize the diffusion model for domain projection in image dehazing. To the best of our knowledge, this is the first time that the diffusion model has been employed to transfer the probability distribution of real-world hazy images (i.e., input) into the synthetic domain (i.e., synthetic hazy). The adapted hazy input is then fed into a pre-trained dehazing model to remove the haze. Consequently, the learned dehazing priors from the synthetic domain can be effectively utilized. In our DiffAD framework, the diffusion bridge serves as a preprocessing step prior to the pre-trained dehazing model. DiffAD provides a novel scheme for real image dehazing and has achieved promising results.
>
> In addition, the custom loss is employed to guide the reverse process of the diffusion, which is also different from previous diffusion bridge and Schrödinger bridge.
>
> ---
>
> **Q2**: The loss functions are heavily handcrafted and yield lots of hyper-parameters, and basically build upon existing loss functions, such as grey-world-assumption loss, dcp loss, etc., which is somewhat less informative.
>
> **A2**: The total loss function used in our DiffAD can be divided into two items: fidelity loss $\mathcal{F}(\tilde{x}\_0, y)$ and quality loss $\mathcal{Q}(\tilde{x}\_0, y)$. Fidelity loss consists of a spatial consistency loss $\mathcal{L}\_{sc}$ and a color consistency loss $\mathcal{L}\_{cc}$. Quality loss consists of a white balance loss $\mathcal{L}\_{wb}$ and a region-aware DCP loss $\mathcal{L}\_{rdcp}$. All the loss functions, except for the spatial consistency loss $\mathcal{L}\_{sc}$ which is adopted from Guo et al. (2020), have been specifically designed for the task of dehazing.
>
> - Color consistency loss $\mathcal{L}\_{cc}$: We propose this novel loss function to deal with the color distortion for our DiffAD. In general circumstances, we don’t need to consider the issue of color distortion, since constraints have been imposed by image distance losses (e.g., MSE). However, in our DiffAD pipeline, MSE may fail the image adaptation ($\tilde{x}\_0$ and $y$ should exhibit distinct distributions). Therefore, $\mathcal{L}\_{cc}$ is designed to encourage color coherence of $H\_{\mathcal{T} \rightarrow \mathcal{S}}$ through preserving the relative color (between channels) between $\tilde{x}\_0$ and $y$. To the best of our knowledge, this is the first time that color consistency loss $\mathcal{L}\_{cc}$ is proposed to align the color information.
> - White balance loss $\mathcal{L}\_{wb}$: We revise the color constancy loss from Guo et al. (2020) and re-name it to white balance loss. It eliminates the color cast of $\tilde{x}\_0$ based on the Gray-World Assumption. Specifically, according to equation 1, regions with dense haze demonstrate increased sensitivity to atmospheric light with color shift. Therefore, we introduce haze density as the spatial weights. Our $\mathcal{L}\_{wb}$ can be regarded as the enhanced version of the color constancy loss.
> - Region-aware DCP loss $\mathcal{L}\_{rdcp}$: DCP loss (Golts et al., 2020; Li et al., 2020) is widely used in real image dehazing. However, DCP tends to fail in the sky region (He et al., 2010). We revise the original DCP loss (Li et al., 2020) and re-name it to region-aware DCP loss $\mathcal{L}\_{rdcp}$.  Accordingly, we exclude the sky region with a mask $\mathcal{M}\_{sky}$ to avoid potential inaccurate calculation of DCP. Our $\mathcal{L}\_{rdcp}$ can be regarded as the comprehensive version of initial DCP loss.
>
> We have also performed ablation study to demonstrate the function of each loss on RTTS dataset. Please refer to Fig. 15 in our revised PDF.
>
> In addition, we have re-organized Sec. 4.1.2 to provide more information about the loss design.

---

> ### Author Response · Authors · 2024-11-24
> **Author Response 2 (highlighted with blue in revised PDF)**
>
> **Q3**: It is recommended to show the captured source distribution in the trained diffusion model for verification. Additionally, according to sec. 5.1, the diffusion model is only trained for 50k iterations from scratch, is this enough for diffusion model to learn the source distribution. Some generated source domain hazy image should be presented and compared with original source image.
>
> **A3**: We utilize the diffusion model trained on ITS and OTS to generate 100 indoor and 100 outdoor hazy images, respectively. We also randomly sample 100 hazy images from the two source domain, respectively. As shown in Fig. 9, even though our diffusion models are only trained for 50k iterations, it has successfully learned the haze distribution in the source domain.
>
> We leverage VGG19 to extract features from these 400 hazy images and apply t-SNE for dimensionality reduction, as shown in  Fig. 9 (e). It can be observed that the generated source domain images are mixed with the original source domain images on the t-SNE map, while hazy images from different source domains remain separated from one another. This further validates that our trained diffusion model can effectively capture the haze distribution of the source domain.
>
> ---
>
> **Q4**: It is unclear when to add the proposed loss functions, are there added in test time as existing zero-shot inverse problem solver, or added in the training time.
>
> **A4**: To avoid confusion, we have re-organized Sec. 4.1 and Fig. 2 to better show the implementation of loss function $\mathcal{L}(\tilde{x}\_0, y)$ (**please refer to the revised PDF**).
>
> The loss function $\mathcal{L}(\tilde{x}\_0, y)$ is embedded in test time to control the reverse process towards the condition $y$ at each time step $t$. It works in test time by guiding the projection from $H\_\mathcal{T}$ to $H\_{\mathcal{T} \rightarrow \mathcal{S}}$.

---

> > ### Author Response · Authors · 2024-11-30
> >
> > Dear Reviewer 9KzP,
> >
> > We would like to express our gratitude for your valuable feedback.
> > We have carefully considered all suggestions and updated our paper accordingly.
> >
> > At present, we are eager to know if we have adequately addressed your questions and concerns.
> > Kindly hope you can help us again in further improving this work.
> >
> > Thank you for your hard work and support.
> >
> > Best,
> >
> > Authors

---

> > ### Comment · Reviewer_9KzP · 2024-12-03
> > **Official Comment by Reviewer 9KzP**
> >
> > Thanks for the authors' response. And the inclusion of the generated source images is appreciated. However, the main concern still exists. Basically, the idea of utilizing diffusion models to bridge two distributions has been heavily presented in literatures, and simply applying it in the dehazing task with real-to-sim distribution translation is somewhat not surprisingly and less informative. Additionally, the proposed method involves massive handcrafted designs, such as loss functions and pseudo label generation, which is kind of like manual patching with less inspiration and sacrifice elegant implementation. Therefore, I will maintain my score.

---

> > > ### Author Response · Authors · 2024-12-03
> > > **Author Response 3**
> > >
> > > Thank you again for your efforts in reviewing our work and for your response.
> > >
> > > We would like to emphasize that, although there have been some works using diffusion models for domain translation, these methods cannot be simply applied to image dehazing tasks without careful desgin. Unlike previous works (e.g., translating a cat image into a tiger image), image restoration tasks like dehazing require us to maintain consistency between the input and output. As shown in Fig. 3, directly applying diffusion-based translation to the input can cause structural distortions and color discrepancies. While the overall dehazing effect seems to be improved, it also introduces additional artifacts, which is unacceptable.
> > >
> > > To address this critical issue, we introduce a conditional generation process and incorporate two fidelity losses, which also needs to be carefully designed according to the characteristics of domain translation. “Absolute” fidelity would lead to the failure of domain translation. Therefore, we propose the spatial consistency loss and color consistency loss to achieve a well trade-off between fidelity and the effect of domain translation.
> > >
> > > Additionally, the reason why we perform input adaptation is because we believe that the model can learn a strong dehazing prior from synthetic datasets. Leveraging well-learned dehazing priors on synthetic dataset is seldom considered in real image dehazing. Reviewer tedF also mentioned that our idea is impressive.
> > >
> > > Finally, each of our designs has a particular purpose. The quality loss allows users to customize and control the dehazing effect (as shown in Fig. 10 and Fig. 11). Pseudo labels help us to train a model that can infer 512x512 images in about 18ms (Tab. 6). These more complex designs provide greater flexibility to users and further simplify the model.
> > >
> > > Following your suggestion, we will continue to seek a more elegant way to improve our framework in future work. Our ultimate goal is to make the most of our pipeline to tap into the full potential of synthetic data, while sidestepping the challenging task of collecting paired hazy and haze-free image pairs from real-world scenes.
> > >
> > > Once again, thank you for your effort, and we sincerely hope you can consider our work.

---

### Official Review · Reviewer_V9fx · 2024-11-03

**Soundness:** 3
**Presentation:** 2
**Contribution:** 3
**Rating:** 6
**Confidence:** 4

**Summary:**

This paper addresses the domain shift issue in single-image dehazing, highlighting the limitations of traditional supervised methods that rely heavily on synthetic data. The authors propose an approach utilizing DDPM to align the input image distributions of the target and source domains. By training the DDPM on haze images from the source domain, the authors can convert target domain images to a format compatible with a pre-trained state-of-the-art dehazing model during testing. Additionally, the authors employ pseudo-labeling to fine-tune the base model, further improving its efficiency.  Finally,  experimental results show that the proposed method outperforms existing dehazing techniques under domain shift conditions in real-world scenes.

**Strengths:**

1. DiffAD adopts an input adaptive approach, which can better utilize the existing dehazing prior knowledge in the source domain without the need to retrain the model. Besides, it introduces an adjustable quality loss function, which allows users to adjust the dehazing and color-balancing effects as needed.
2. DiffAD also provides a method to generate high-quality pseudo-labels to improve the generalization ability of the model, experiments verify the effectiveness of the proposed method in real-world image dehazing.

**Weaknesses:**

1. The proposed method requires training a denoising diffusion probability model as well as a state-of-the-art dehazing model, which makes the whole method very complex.
2. The datasets used in this method are all synthetic or limited datasets, which may lead to poor performance in dealing with real-world dehazing tasks，therefore, authors need to provide more results on other datasets.
3.  The authors only provide non-reference metrics to evaluate the performance, however, MOS-based metrics should be provided in the paper, since non-reference metrics can not fully reveal the human preference.
4.  Since the proposed method seems to be effective in real-world image dehazing, can it be extended to other tasks, such as real image super-resolution or real-world image denoising?

**Questions:**

Please see above weakness.

---

> ### Author Response · Authors · 2024-11-24
> **Author Response 1 (highlighted with green in revised PDF)**
>
> **Q1**: The proposed method requires training a denoising diffusion probability model as well as a state-of-the-art dehazing model, which makes the whole method very complex.
>
> **A1**: We only need to train a single diffusion model to learn the distribution of hazy images (for each source domain). Once such a diffusion model is trained, it can be applied to dehazing models trained on the same source domain to improve their generalization ability.
>
> In practical applications, there is no need to re-train a state-of-the-art dehazing model. Instead, the trained diffusion model can be directly applied to pre-trained dehazing models provided in the open-source community.
>
> Additionally, we further propose DiffAD-S-FT to avoid the potential complexity caused by the diffusion model (Sec 4.2). As shown in Table 6, for a 512×512 image, DiffAD-S-FT only takes about 18ms to generate a clear dehazed image, achieving good trade-off between performance and efficiency.
>
> ---
>
> **Q2**: The datasets used in this method are all synthetic or limited datasets, which may lead to poor performance in dealing with real-world dehazing tasks, therefore, authors need to provide more results on other datasets.
>
> **A2:** In Sec. 5.3 of our manuscript, we have discussed the real-world dehazing performance of our DiffAD-FT and conduct both quantitative and qualitative comparisons with other state-of-the-art real dehazing methods. Experimental results demonstrate that our DiffAD-FT is effective in real-world dehazing. We also provide additional visual results on Fattal’s dataset in Fig. 20 of appendix.
>
> We further adopt the RTTS dataset to evaluate the effectiveness of directly applying our DiffAD to the pre-trained dehazing models. Specifically, we select models trained on ITS and OTS to test their performance on RTTS dataset. RTTS dataset contains 4,322 images of various real-world hazy scenes, making it an effective dataset for assessing real-world dehazing performance. Since the RTTS dataset lacks ground truths, we use BRISQUE [1] and PaQ-2-PiQ [2] as no-reference evaluation metrics. The quantitative results are summarized in the following table.
>
> | Settings                   | ITS/RTTS | ITS/RTTS  | OTS/RTTS | OTS/RTTS  |
> |----------------------------|----------|-----------|----------|-----------|
> |                            | BRISQUE$\downarrow$  | PaQ-2-PiQ$\uparrow$ | BRISQUE$\downarrow$  | PaQ-2-PiQ$\uparrow$ |
> | (CVPR'21) AECRNet          | 29.40    | 66.30     | 30.64    | 66.56     |
> | (Ours) AECRNet-DiffAD      | **28.43**    | **67.69**     | **28.83**    | **66.79**     |
> | (TIP'23) Dehazeformer      | 30.17    | 66.61     | 33.16    | 66.60     |
> | (Ours) Dehazeformer-DiffAD | **29.49**    | **67.38**     | **30.10**    | **66.78**     |
> | (ICCV'23) FocalNet         | 33.75    | 66.35     | 35.82    | 66.48     |
> | (Ours) FocalNet-DiffAD     | **31.53**    | **67.48**     | **32.15**    | **66.93**     |
>
> As shown, after using DiffAD, both no-reference evaluation metrics improved, indicating that the image quality of the dehazing results was effectively enhanced due to the input adaptation performed by our DiffAD.
>
> We also present qualitative results in Fig. 14.  It can be observed that due to the significant domain gap, the dehazing models pre-trained on both ITS and OTS datasets demonstrate limited dehazing ability, leaving a considerable amount of haze residue in the dehazed images. However, with DiffAD, the pre-trained dehazing model can correctly leverage the dehazing priors learned from synthetic datasets, substantially improving the dehazing performance.
>
> [1] Mittal, Anish, et al. “No-reference image quality assessment in the spatial domain.” IEEE TIP 2012.
>
> [2] Ying, Zhenqiang, et al. "From patches to pictures (PaQ-2-PiQ): Mapping the perceptual space of picture quality." CVPR 2020.
>
> ---
>
> **Q3**: The authors only provide non-reference metrics to evaluate the performance, however, MOS-based metrics should be provided in the paper, since non-reference metrics can not fully reveal the human preference.
>
> **A3**: Based on the suggestion of this reviewer, we conduct a user study on the RTTS dataset. Specifically, we randomly select 50 hazy images from the RTTS dataset and compare our results with four competitors (i.e., DAD, PSD, D4 and RIDCP). We invite 12 experts with experience in image processing to choose the best method. Participants are asked to consider both the presence of residual haze and artifacts caused by over-dehazing when making their decisions. We utilize the Dell S2721DGF monitor to display images. Finally, we statistic the selection ratios for each method in the following table. We observe that user study results align with the no-reference evaluation, indicating that our method is the most favored one.
>
> |            | DAD  | PSD  | D4   | RIDCP | Ours |
> |------------|------|------|------|-------|------|
> | Percentage | 0.06 | 0.03 | 0.04 | 0.32  | 0.55 |

---

> ### Author Response · Authors · 2024-11-24
> **Author Response 2 (highlighted with green in revised PDF)**
>
> **Q4**: Since the proposed method seems to be effective in real-world image dehazing, can it be extended to other tasks, such as real image super-resolution or real-world image denoising?
>
> **A4**: Thank you for providing this insightful comment. Our DiffAD pipeline can be extended to real image super-resolution/real-world image denoising by adapting the real low-resolution image/real-world noisy image to bicubic/Gaussian domain. The custom loss function should be re-designed to fit these low-level tasks for state-of-the-art performance. Due to the limited time, we leave it for our future work.

---

> > ### Author Response · Authors · 2024-11-30
> >
> > Dear Reviewer V9fx,
> >
> > We would like to express our gratitude for your valuable feedback.
> > We have carefully considered all suggestions and updated our paper accordingly.
> >
> > At present, we are eager to know if we have adequately addressed your questions and concerns.
> > Kindly hope you can help us again in further improving this work.
> >
> > Thank you for your hard work and support.
> >
> > Best,
> >
> > Authors

---

> ### Comment · Reviewer_V9fx · 2024-11-30
>
> Thanks for your rebuttal. In my mind, I highly appreciate the point of integrating the diffusion model to alleviate the domain shift issue, although I would prefer to integrate the mechanism of diffusion to solve this issue implicitly, this is why I raise the point "which makes the whole method very complex" in the first issue. In my mind, the design of the proposed method is not elegant enough, and on the other hand, the mind of this paper towards generalized image restoration by diffusion mechanism is on the right way. Moreover, the issue that I raise "can it be extended to other tasks, such as real image super-resolution or real-world image denoising" is that the domain shift issues in dehazing, denoising, and SR are significantly different, a discussion of these problems would encourage more research towards designing a general approach using diffusion mechanism for generalized image reconstruction tasks. If possible, I hope the authors can discuss this point. Considering the rebuttal and other reviewers' comments, I would like to keep a positive score and raise my confidence currently.

---

> > ### Author Response · Authors · 2024-12-02
> > **Author Response 3**
> >
> > Thank you very much for your patient response and the effort you have dedicated to reviewing our work. We have been considering the possibility of extending our DiffAD to other image restoration / enhancement tasks. We have attempted to apply our DiffAD to other tasks, and would like to share some of our thoughts during this process for further discussion:
> >
> > 1. As you pointed out, the domain shift issue varies significantly across different tasks. For tasks such as super-resolution and denoising, the pre-trained models focus more on recovering high-frequency information. Our DiffAD, however, is designed for image dehazing, where recovering low-frequency information (e.g., brightness and contrast) is more important. This may explain the significant differences and the challenge of directly applying our DiffAD to achieve SOTA performance in other tasks.
> > 2. We design different quality losses (e.g., $\mathcal{L}\_{rdcp}$ and $\mathcal{L}\_{wb}$) for DiffAD specifically for image dehazing. As shown in Tab. 3 and Fig. 15 in the revised PDF, the quality loss can significantly improve the dehazing performance. Therefore, we believe that a task-specific quality loss needs to be designed for different tasks. The essence of the quality loss is to provide the pipeline with a reliable reference (e.g., $\mathcal{L}\_{rdcp}$, which essentially provides a DCP result as a reference). In denoising, the reference could be the pre-denoised result of an edge-preserving filter, such as the side window filter [1]. For low-light enhancement, exposure loss [2] can be used as the quality loss. Similarly, for underwater image enhancement, the white balance loss $\mathcal{L}_{wb}$ may be suitable.
> > 3. In addition to designing task-specific quality losses, we also believe that robust no-reference image quality assessment (NRIQA) methods can be used as a unified quality loss. Recent related work is [3].
> > 4. For high-frequency sensitive tasks like super-resolution and denoising, when degradation is particularly severe, leveraging the generative priors learned on high-quality datasets may be another research direction for restoring fine details. Recent works [4][5][6] have validated this idea well.
> >
> > At last, we would like to express our sincere gratitude once again for your suggestions, which have inspired us to think more deeply about these issues. We are open to any further discussions. In future work, we will continue to explore the application of diffusion models to address low-level visual domain shift issues. We aim to expand our focus beyond image dehazing to other tasks, hoping to identify connections between different low-level visual tasks.
> >
> > [1] Yin, Hui, et at. “Side Window Filtering.” CVPR 2019.
> >
> > [2] Guo, Chunle, et al. “Zero-Reference Deep Curve Estimation for Low-Light Image Enhancement.” CVPR 2020.
> >
> > [3] Zhang, Weixia, et al. “Comparison of No-Reference Image Quality Models via MAP Estimation in Diffusion Latents.” arXiv, 2024.
> >
> > [4] Wang, Jianyi, et al. “Exploiting Diffusion Prior for Real-World Image Super-Resolution." IJCV, 2024.
> >
> > [5] Lin, Xinqi, et al. “DiffBIR: Towards Blind Image Restoration with Generative Diffusion Prior." ECCV, 2024.
> >
> > [6] Yu, Fanghua, et al. “Scaling Up to Excellence: Practicing Model Scaling for Photo-Realistic Image Restoration In the Wild.” CVPR 2024.

---

### Note · Authors · 2025-03-08

I have read and agree with the venue's withdrawal policy on behalf of myself and my co-authors.

---

### Meta-Review · Area_Chair_1zzE · 2024-12-20

**Metareview:**

The manuscript proposes a diffusion model-based adaptation method to address the domain shift between synthetic and real-world hazy scenarios. The approach leverages a denoising diffusion probabilistic model trained on source hazy images to capture the prior probability distribution of the source domain, augmenting state-of-the-art dehazing methods with synthetic hazy images.
The paper received five reviews, with three recommending acceptance and two recommending rejection. Reviewers raised several critical concerns, including limited contributions, inadequate benchmarks, biased evaluation metrics, lack of generalization to other tasks, insufficient ablation studies on loss functions, a lack of qualitative results, and compositional issues. Despite efforts made during the rebuttal stage, some concerns remain unresolved, and the manuscript does not meet the acceptance threshold for this venue.

**Additional Comments On Reviewer Discussion:**

Prior to the rebuttal phase, three reviewers (V9fx, yP7K, and tedF) rated the paper "6 - marginally above the acceptance threshold," while two reviewers (9KzP and fxe7) rated it "5 - marginally below the acceptance threshold." Post-rebuttal, V9fx and yP7K maintained their ratings of 6, whereas 9KzP upheld their rejection rating, citing persistent concerns regarding limited contributions and extensive reliance on handcrafted designs and loss functions.

---

### Decision · Program_Chairs · 2025-01-22

Reject